# Robust Gaussian Processes via Relevance Pursuit

**Sebastian Ament**
Meta
ament@meta.com

**Elizabeth Santorella**
Meta
santorella@meta.com

**David Eriksson**
Meta
deriksson@meta.com

**Ben Letham**
Meta
bletham@meta.com

**Maximilian Balandat**
Meta
balandat@meta.com

**Eytan Bakshy**
Meta
ebakshy@meta.com

## Abstract

Gaussian processes (GPs) are non-parametric probabilistic regression models that are popular due to their flexibility, data efficiency, and well-calibrated uncertainty estimates. However, standard GP models assume homoskedastic Gaussian noise, while many real-world applications are subject to non-Gaussian corruptions. Variants of GPs that are more robust to alternative noise models have been proposed, and entail significant trade-offs between accuracy and robustness, and between computational requirements and theoretical guarantees. In this work, we propose and study a GP model that achieves robustness against sparse outliers by inferring data-point-specific noise levels with a sequential selection procedure maximizing the log marginal likelihood that we refer to as *relevance pursuit*. We show, surprisingly, that the model can be parameterized such that the associated log marginal likelihood is *strongly concave* in the data-point-specific noise variances, a property rarely found in either robust regression objectives or GP marginal likelihoods. This in turn implies the weak submodularity of the corresponding subset selection problem, and thereby proves approximation guarantees for the proposed algorithm. We compare the model's performance relative to other approaches on diverse regression and Bayesian optimization tasks, including the challenging but common setting of sparse corruptions of the labels within or close to the function range.

## 1 Introduction

Probabilistic models have long been a central part of machine learning, and Gaussian process (GP) models are a key workhorse for many important tasks [54], especially in the small-data regime. GPs are flexible, non-parametric predictive models known for their high data efficiency and well-calibrated uncertainty estimates, making them a popular choice for regression, uncertainty quantification, and downstream applications such as Bayesian optimization (BO) [8, 24, 26] and active learning [6, 55].

GPs flexibly model a distribution over functions, but assume a particular observation model. The standard formulation assumes i.i.d Gaussian observation noise, i.e., $y(\mathbf{x}) = f(\mathbf{x}) + \epsilon$, where $f(\mathbf{x})$ is the true (latent) function value at a point $\mathbf{x}$ and $\epsilon \sim \mathcal{N}(0, \sigma^2)$, implying a homoskedastic Gaussian likelihood. While mathematically convenient, this assumption can be a limitation in practice, since noise distributions are often heavy-tailed or observations may be corrupted due to issues such as sensor failures, data processing errors, or software bugs. Using a standard GP model in such settings can result in poor predictive performance.

A number of *robust* GP modeling approaches have been proposed to remedy this shortcoming, most of which fall into the following broad categories: data pre-processing (e.g., Winsorizing), modified likelihood functions (e.g., Student-$t$), and model-based data selection and down-weighting procedures.

38th Conference on Neural Information Processing Systems (NeurIPS 2024).

These approaches offer different trade-offs between model accuracy, degree of robustness, broad applicability, computational requirements, and theoretical guarantees.

In this paper, we propose a simple yet effective implicit data-weighting approach that endows GPs with a high degree of robustness to challenging label corruptions. Our approach is flexible and can be used with arbitrary kernels, is efficient to compute, and yields provable approximation guarantees. Our main contributions are as follows:

1. We propose a modification to the standard GP model that introduces learnable data-point-specific noise variances.

2. We introduce a novel greedy sequential selection procedure for maximizing the model's marginal log-likelihood (MLL) that we refer to as *relevance pursuit*.

3. We prove that, under a particular parameterization, the MLL is strongly concave in the data-point-specific noise variances, and derive approximation guarantees for our algorithm.

4. We demonstrate that our approach, Robust Gaussian Processes via Relevance Pursuit (RRP), performs favorably compared to alternative methods across various benchmarks, including challenging settings of sparse label corruptions within the function's range, see e.g. Figure 1.

## 2 Preliminaries

We aim to model a function $f : \mathbb{X} \to \mathbb{R}$ over some domain $\mathbb{X} \subset \mathbb{R}^d$. With a standard Gaussian noise model, for $\mathbf{x}_i \in \mathbb{X}$ we obtain observations $y_i = f(\mathbf{x}_i) + \epsilon_i$, where $\epsilon_i \sim \mathcal{N}(0, \sigma^2)$ are i.i.d. draws from a Gaussian random variable. $\|\cdot\|$ denotes the Euclidean norm unless indicated otherwise.

### 2.1 Gaussian Processes

A GP $f \sim \mathcal{GP}(\mu(\cdot), k_{\boldsymbol{\theta}}(\cdot, \cdot))$ is fully defined by its mean function $\mu : \mathbb{X} \to \mathbb{R}$ and covariance or kernel function $k_{\boldsymbol{\theta}} : \mathbb{X} \times \mathbb{X} \to \mathbb{R}$, which is parameterized by $\boldsymbol{\theta}$. Without loss of generality, we will assume that $\mu \equiv 0$. Suppose we have collected data $\mathcal{D} = \{(\mathbf{x}_i, y_i)\}_{i=1}^n$ where $\mathbf{X} := \{\mathbf{x}_i\}_{i=1}^n$, $\mathbf{y} := \{y_i\}_{i=1}^n$. Let $\boldsymbol{\Sigma}_{\boldsymbol{\theta}} \in \mathcal{S}_{++}^n$ denote the covariance matrix of the data set, i.e., $[\boldsymbol{\Sigma}_{\boldsymbol{\theta}}]_{ij} = k_{\boldsymbol{\theta}}(\mathbf{x}_i, \mathbf{x}_j) + \delta_{ij}\sigma^2$, where $\delta_{ij}$ is the Kronecker delta. The negative marginal log-likelihood (NMLL) $\mathcal{L}$ is given by

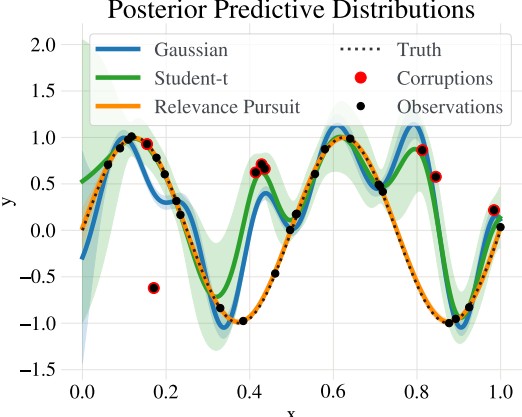

Figure 1: Comparison of RRP to a standard GP and a variational GP with a Student-$t$ likelihood on a regression example. While the other models are led astray by the corrupted observations, RRP successfully identifies the corruptions (red) and thus achieves a much better fit to the ground truth.

$$-2\mathcal{L}(\boldsymbol{\theta}) := -2\log p(\mathbf{y}|\mathbf{X}, \boldsymbol{\theta}) = \mathbf{y}^\top \boldsymbol{\Sigma}_{\boldsymbol{\theta}}^{-1}\mathbf{y} + \log\det\boldsymbol{\Sigma}_{\boldsymbol{\theta}} + n\log 2\pi. \quad (1)$$

In the following, we will suppress the explicit dependence of the kernel matrix on $\boldsymbol{\theta}$ for brevity of notation. For a comprehensive background on GPs, we refer to Rasmussen et al. [54].

### 2.2 Noise Models

**Additive, heavy-tailed noise** Instead of assuming the noise term $\epsilon_i$ in the observation model to be Gaussian, other noise models consider zero-mean perturbations drawn from distributions with heavier tails, such as the Student-$t$ [32], Laplace [39], or $\alpha$-Stable [5] distributions. These types of errors are common in applications such as finance, geophysics, and epidemiology [18]. Robust regression models utilizing Student-$t$ errors are commonly used to combat heavy-tailed noise and outliers.

**Sparse corruptions** In practice, often a small number of labels are corrupted. We will refer to these as "outliers," though emphasize that the corrupted values may fall within the range of normal outputs. Sparse corruptions are captured by a model of the form $y_i = Z_i f(\mathbf{x}_i) + (1 - Z_i)W_i$, where

$Z_i \in \{0, 1\}$ and $W_i \in \mathbb{R}$ is a random variable. Note that $W_i$ need not have (and rarely has) $f(\mathbf{x}_i)$ as its mean. For instance, consider a faulty sensor that with some probability $p$ reports a random value within the sensor range $[y_l, y_h]$. In this case $Z_i \sim \text{Ber}(p)$ and $W_i \sim \text{U}[y_l, y_h]$. Software bugs, such as those found in ML training procedures, or errors in logging data can result in sparse corruptions.

## 3   Related Work

**Data pre-processing**   Data pre-processing can be an effective technique for handling simple forms of data corruption, such as values that fall outside a valid range of outputs. With such pre-processing, outliers are handled upstream of the regression model. Common techniques include the power transformations [15], trimming, and winsorization. These methods can add substantial bias if not used carefully, and generally do not handle data corruptions that occur within the normal range of the process to be modeled. See [17] for a review on data cleaning.

**Heavy-tailed likelihoods**   One class of robust methods uses additive heavy-tailed noise likelihoods for GPs, particularly Student-$t$ [32], Laplace [53], and Huber [1], and could be extended with $\alpha$-Stable distributions, which follow a generalized central limit theorem [5]. These models are less sensitive to outliers, but they lose efficiency when the outliers are a sparse subset of the observations, as opposed to global heavy-tailed noise. Furthermore, model inference is no longer analytic, necessitating the use of approximate inference approaches such as MCMC [49], Laplace approximation [64], expectation propagation (EP) [32], Expectation Maximization [53], or variational inference [61]. Shah et al. [56] take a related approach using a Student-$t$ process prior in the place of the GP prior. Unfortunately, the Student-$t$ process is not closed under addition and lacks the tractability that makes GPs so versatile. Alternative noise specifications include a hierarchical mixture of Gaussians [19] and a "twinned" GP model [48] that uses a two-component noise model to allow outlier behavior to depend on the inputs. This method is suited for settings where outliers are not totally stochastic, but generally is not able to differentiate "inliers" from outliers when they can occur with similar inputs.

**Outlier classification**   Awasthi et al. [11] introduces the Trimmed MLE approach, which identifies the subset of data points (of pre-specified size) under which the marginal likelihood is maximized. Andrade and Takeda [10] fit GPs using the trimmed MLE by applying a projected gradient method to an approximation of the marginal likelihood. The associated theory only guarantees convergence to a stationary point, with no guarantee on quality. When no outliers are present, this method can be worse than a standard GP. Li et al. [40] propose a heuristic iterative procedure of removing those data points with the largest residuals after fitting a standard GP, with subsequent reweighting. The method shows favorable empirical performance but has no theoretical guarantees, and fails if the largest residual is not associated with an outlier. Park et al. [52] consider a model of the form $y_i = \delta_i + f(\mathbf{x}_i) + \epsilon_i$, where outliers are regarded as data with a large bias $\delta_i$. Their random bias model is related to our model in that it also introduces learnable, data-point-specific variances. However, inference is done in one step by optimizing the NMLL with an inverse-gamma prior on the $\delta_i$'s, which – in contrast to the method proposed herein – generally does not lead to exactly sparse $\delta_i$'s .

**Sample re-weighting**   Altamirano et al. [2] propose robust and conjugate GPs (RCGP) based on a modification to the Gaussian likelihood function that is equivalent to standard GP inference, where the covariance of the noise $\sigma^2 \mathbf{I}$ is replaced by $\sigma^2 \text{diag}(\mathbf{w}^{-2})$ and the prior mean $\mathbf{m}$ is replaced by $\mathbf{m_w} = \mathbf{m} + \sigma^2 \nabla_y \log(\mathbf{w}^2)$. The authors advocate for the use of the inverse multi-quadratic weight function $w(\mathbf{x}, y) = \beta(1 + (y - m(\mathbf{x}))^2/c^2)^{-1/2}$, which introduces two additional hyper-parameters: the soft threshold $c$, and the "learning rate" $\beta$. Importantly, the weights $\mathbf{w}$ are defined *a-priori* as a function of the prior mean $m(\mathbf{x})$ and the targets $y$, thereby necessitating the weights to identify the correct outliers without access to a model. This is generally only realistic if the outlier data points are clearly separated in the input or output spaces rather than randomly interspersed.

## 4   Robust Gaussian Process Regression via Relevance Pursuit

Our method adaptively identifies a sparse set of outlying data points that are corrupted by a mechanism that is not captured by the other components of the model. This is in contrast to many other approaches to robust regression that non-adaptively apply a heavy-tailed likelihood to *all* observations, which can be suboptimal if many observations are of high quality.

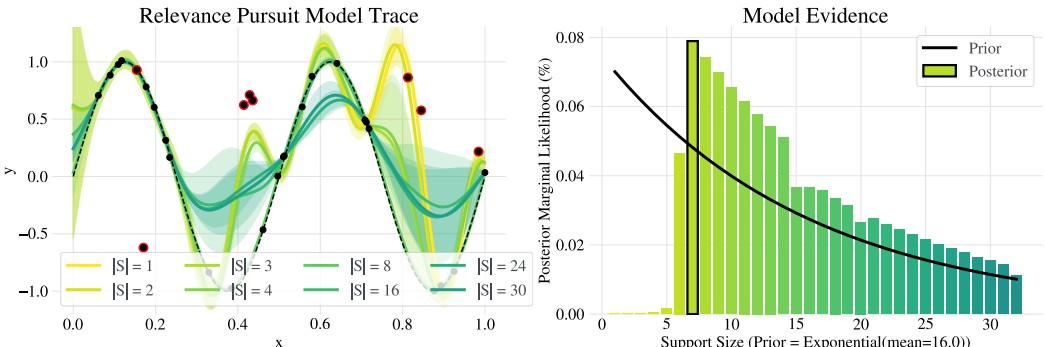

Figure 2: *Left:* Evolution of model posterior during Relevance Pursuit, as the number of data-point-specific variances $|S|$ increases (from light colors to dark). Red points indicate corruptions that were generated by uniformly sampling from the function's range. *Right:* Comparison of posterior marginal likelihoods as a function of a model's $|S|$. The maximizer – boxed in black – is the preferred model.

### 4.1 The Extended Likelihood Model

We extend the standard GP observation noise variance $\sigma^2$ with data-point-specific noise variances $\boldsymbol{\rho} = \{\rho_i\}_{i=1}^n$, so that the $i$-th data point is distributed as

$$y_i \mid \mathbf{x}_i \sim \mathcal{N}\left(f(\mathbf{x}_i), \sigma^2 + \rho_i\right). \tag{2}$$

This is similar to Sparse Bayesian Learning [60] in which weight-specific prior variances control a feature's degree of influence on a model's predictions. The marginal likelihood optimization of $\rho_i$ in (2) gives rise to an *automatic mechanism* for the detection and weighting of outliers. The effect of $y_i$ on the estimate of $f$ vanishes as $\rho_i \to \infty$, similar to the effect of the latent varibales $\mathbf{h}$ in Bodin et al. [14]'s extended GP model $f(\mathbf{x}, \mathbf{h})$, though $\mathbf{h}$ requires MCMC for inference. While many heteroskedastic GP likelihoods model noise as an input-dependent process [28, 35], our formulation does not require such assumptions, and is thus suitable for corruptions that are not spatially correlated.

An elegant consequence of our modeling assumption is that we can compute individual marginal-likelihood maximizing $\rho_i$'s in closed form when keeping all $\rho_j$ for $j \neq i$ fixed. In particular,

**Lemma 1.** *[Optimal Robust Variances] Let $\mathcal{D}_{\backslash i} = \{(\mathbf{x}_j, y_j) : j \neq i\}$, $\boldsymbol{\rho} = \boldsymbol{\rho}_{\backslash i} + \rho_i \mathbf{e}_i$, where $\boldsymbol{\rho}, \boldsymbol{\rho}_{\backslash i} \in \mathbb{R}_+^n$, $[\boldsymbol{\rho}_{\backslash i}]_i = 0$, and $\mathbf{e}_i$ is the $i$th canonical basis vector. Then keeping $\boldsymbol{\rho}_{\backslash i}$ fixed,*

$$\rho_i^* = \arg\max_{\rho_i} \mathcal{L}\left(\boldsymbol{\rho}_{\backslash i} + \rho_i \mathbf{e}_i\right) = \left[(y_i - \mathbb{E}[y(\mathbf{x}_i)|\mathcal{D}_{\backslash i}])^2 - \mathbb{V}[y(\mathbf{x}_i)|\mathcal{D}_{\backslash i}]\right]_+, \tag{3}$$

*where $y(\mathbf{x}_i) = f(\mathbf{x}_i) + \epsilon_i$. These quantities can be expressed as functions of $\boldsymbol{\Sigma}^{-1} = (\mathbf{K} + \mathbf{D}_{\sigma^2 + \boldsymbol{\rho}})^{-1}$:*

$$\mathbb{E}[y(\mathbf{x}_i)|\mathcal{D}_{\backslash i}]^2 = y_i - \left[\boldsymbol{\Sigma}^{-1}\mathbf{y}\right]_i \big/ \left[\boldsymbol{\Sigma}^{-1}\right]_{ii}, \qquad \text{and} \qquad \mathbb{V}[y(\mathbf{x}_i)|\mathcal{D}_{\backslash i}] = 1 \big/ \left[\boldsymbol{\Sigma}^{-1}\right]_{ii},$$

*where $\mathbf{D}_{\sigma^2 + \boldsymbol{\rho}}$ is a diagonal matrix whose entries are $\sigma^2 + \boldsymbol{\rho}$.*

The first component $\mathbb{E}[f(\mathbf{x}_i) + \epsilon_i|\mathcal{D}_{\backslash i}]^2$ of (3) is the empirical error to $y_i$ of the model trained without the $i$-th data point, i.e., the leave-one-out (LOO) cross-validation error [54]. The second component $\mathbb{V}[f(\mathbf{x}_i) + \epsilon_i|\mathcal{D}_{\backslash i}]$ is the LOO predictive variance. The optimal solution to $\rho_i$ is only non-zero for those observations whose squared LOO error is larger than the LOO predictive variance at that point.

### 4.2 Optimization with a Maximum Number of Outliers

Without additional structure, inference of the noise variances $\rho_i$ does not yield desirable models, as the marginal likelihood can be improved by increasing the prior variance $\rho_i$ of any data point where Eq. (3) is greater than zero, even if that is due to regular (non-outlier) measurement noise. To avoid this, we constrain the number of non-zero $\rho_i$, that is, $\|\boldsymbol{\rho}\|_\infty = |\{0 < \rho_i\}| \leq k < n$. While this sparsity constraint mitigates over-flexibility, it gives rise to a formidably challenging optimization problem, as there are a combinatorial number of sparse outlier sets to consider. Even if the number of outliers $n_o$ were known, exhaustive search would still require considering $n$-choose-$n_o$ possibilities.

For tractability, we iteratively add data points to a set of potential "outliers" by setting their associated $\rho_i$ to be nonzero, using the closed-form expression for the optimal individual $\rho_i$ variances in Lemma 1. As the algorithm seeks to identify the most "relevant" data points (as measured by $\mathcal{L}$) upon completion, we refer to it as *Relevance Pursuit*. This is Algorithm 1 with useBayesianModelSelection as false. Specifically, this is the "forward" variant; Algorithm 2 in the Appendix presents an alternative "backward" variant that we found to work well if the number of corrupted data points is large.

Crucial to the performance of the optimizer, it never removes data from consideration completely; a data point is only down-weighted if it is apparently an outlier. This allows the down-weighting to be reversed if a data point appears "inlying" after having down-weighted other data points, improving the method's robustness and performance. This is in contrast to Andrade and Takeda [10]'s greedy algorithm, in which the exclusion of a data point can both increase or decrease the associated marginal likelihood. This means that their objective is not monotonic, a necessary condition to provide constant-factor submodular approximation guarantees for greedy algorithms, see Section 5.

---

**Algorithm 1** Relevance Pursuit (Forward Algorithm)

---

**Require:** $\mathbf{X}$, $\mathbf{y}$, schedule $\mathcal{K} = (k_1, k_2, \ldots, k_{\mathcal{K}})$, useBayesianModelSelection (boolean)

  Initialize $\mathcal{S}_0 \subseteq \{1, \ldots, n\}$ (typically $\mathcal{S}_0 = \emptyset$)

  **for** $i$ in $(1, \ldots, |\mathcal{K}|)$ **do**

    Optimize MLL: $\boldsymbol{\rho}_{\mathcal{S}_i} \leftarrow \arg\max_{\boldsymbol{\rho}_{\mathcal{S}_i}} \mathcal{L}\left(\boldsymbol{\rho}_{\mathcal{S}_i}\right)$,  where $\boldsymbol{\rho}_{\mathcal{S}_i} = \{\boldsymbol{\rho} : \rho_j = 0, \ \forall j \notin \mathcal{S}_i\}$.

    Expand Support:

      Compute $\Delta_i(j) \leftarrow \max_{\rho_j} \mathcal{L}(\boldsymbol{\rho}_{\mathcal{S}_i} + \rho_j \mathbf{e}_j) - \mathcal{L}(\boldsymbol{\rho}_{\mathcal{S}_i})$ for all $j \notin \mathcal{S}_i$ via Lemma 1 .

      $\mathcal{A}_i \leftarrow \{j_1, \ldots, j_{k_i}\}$ such that $\Delta_i(j) \geq \Delta_i(j')$ for all $j \in \mathcal{A}_i$ and $j' \notin (\mathcal{A}_i \cup \mathcal{S}_i)$.

      $\mathcal{S}_{i+1} \leftarrow \mathcal{S}_i \cup \mathcal{A}_i$

  **if** useBayesianModelSelection **then**

    Compute the marginal likelihood $p(\mathcal{D}|\mathcal{S}_i) \approx p(\mathcal{D}|\mathcal{S}_i, \boldsymbol{\rho}_{\mathcal{S}_i})$

    $\mathcal{S}^* \leftarrow \arg\max_{\mathcal{S}_i} p(\mathcal{D}|\mathcal{S}_i) p(\mathcal{S}_i)$.

  **else**

    $\mathcal{S}^* = \mathcal{S}_{\mathcal{K}}$.

  Return $\mathcal{S}^*$, $\boldsymbol{\rho}_{\mathcal{S}^*}$.

---

### 4.3 Automatic Outlier Detection via Bayesian Model Selection

In practice, it is often impossible to set a hard threshold on the number of outliers for a particular data set. For example, a sensor might have a known failure rate, but how many outliers it produces will depend on the specific application of the sensor. Thus, is often more natural to specify a prior distribution $p(\mathcal{S})$ over the number of outliers, rather than fix the number *a priori*. We leverage the Bayesian model selection framework [66, 45] to determine the most probable number of outliers in a data- and model-dependent way, aiming to maximize $p(\mathcal{S}|\mathcal{D})$. This gives rise to Algorithm 1, with useBayesianModelSelction as true.

Computationally, we start by iteratively adding outliers up to the maximal support of the prior, similar to the procedure described in Section 4.2. We store a trace of models generated at each iteration, then approximate the model posterior $p(\mathcal{S}_i|\mathcal{D}) \propto p(\mathcal{D}|S_i) p(S_i)$ at each point in the trace. As the exact posterior is intractable, we approximate it with $p(\mathcal{D}|\mathcal{S}_i) = \int p(\mathcal{D}|\mathcal{S}_i, \boldsymbol{\rho}_{\mathcal{S}_i}) \mathrm{d}\boldsymbol{\rho}_{\mathcal{S}_i} \approx p(\mathcal{D}|\mathcal{S}_i, \boldsymbol{\rho}_{\mathcal{S}_i}^*)$. Finally, we select the model from the model trace $\{\mathcal{S}_i\}_i$ that attains the highest model posterior likelihood. Imposing a prior on the number of outliers differs notably from most sparsity-inducing priors, which are instead defined on the parameter values, like $l_1$-norm regularization. In practice, $p(\mathcal{S})$ can be informed by empirical distributions of outliers. For our experiments, we use an exponential prior on $|\mathcal{S}|$ to encourage the selection of models that fit as much of the data as tightly as possible.

Regarding the schedule $\mathcal{K}$ in Algorithm 1, the most natural choice is simply to add one data point at a time, i.e. $\mathcal{K} = (1, 1, ...)$, but this can be slow for large $n$. In practice, we recommend schedules that test a fixed set of outlier fractions, e.g. $\mathcal{K} = (0.05n, 0.05n, \ldots)$.

## 5 Theoretical Analysis

We now provide a theoretical analysis of our approach. We first propose a re-parameterization of the $\rho_i$ that maps the optimization problem to a compact domain. Surprisingly, the re-parameterized

problem exhibits strong convexity and smoothness when the base covariance matrix (excluding the $\rho_i$) is well-conditioned. We connect the convexity and smoothness with existing results that yield approximation guarantees for sequential greedy algorithms, implying a constant-factor approximation guarantee to the optimal achievable NMLL value for generalized orthogonal matching pursuit (OMP), a greedy algorithm that is closely related to Algorithm 1.

## 5.1 Preliminaries for Sparse Optimization

The optimization of linear models with respect to least-squares objectives in the presence of sparsity constraints has been richly studied in statistics [59], compressed sensing [3, 62], and machine learning [9, 67]. Of central importance to the theoretical study of this problem class are the eigenvalues of sub-matrices of the feature matrix, corresponding to sparse feature selections and so often referred to as *sparse eigenvalues*. The restricted isometry property (RIP) formalizes this.

**Definition 2** (Restricted Isometry Property). *An $(n \times m)$-matrix $\mathbf{A}$ satisfies the $r$-restricted isometry property (RIP) with constant $\delta_r \in (0, 1)$ if for every submatrix $\mathbf{A}_\mathcal{S}$ with $|\mathcal{S}| = r \leq m$ columns,*

$$(1 - \delta_r)\|\mathbf{x}\| \leq \|\mathbf{A}_\mathcal{S}\mathbf{x}_\mathcal{S}\| \leq (1 + \delta_r)\|\mathbf{x}\|,$$

*where $\mathbf{x}_\mathcal{S} \in \mathbb{R}^r$. This is equivalent to $(1 - \delta_r)\mathbf{I} \preceq (\mathbf{A}_\mathcal{S}^*\mathbf{A}_\mathcal{S}) \preceq (1 - \delta_r)\mathbf{I}$.*

The RIP has been proven to lead to exact recovery guarantees [16], as well as approximation guarantees [20]. Elenberg et al. [23] generalized the RIP to non-linear models and other data likelihoods, using the notion of restricted strong convexity (RSC) and restricted smoothness.

**Definition 3** (Restricted Strong Convexity and Smoothness). *A function $f : \mathbb{R}^d \to \mathbb{R}$ is $m_r$-restricted strong convex and $M_r$-restricted smooth if for all $(\mathbf{x}, \mathbf{x}')$ in the domain $D_r \subset (\mathbb{R}^d \times \mathbb{R}^d)$,*

$$m_r\|\mathbf{x}' - \mathbf{x}\|^2/2 \;\leq\; f(\mathbf{x}') - f(\mathbf{x}) - \nabla[f](\mathbf{x})^\top(\mathbf{x}' - \mathbf{x}) \;\leq\; M_r\|\mathbf{x}' - \mathbf{x}\|^2/2.$$

*In the context of sparse optimization, we let $D_r$ be the set of tuples of $r$-sparse vectors whose difference is also at most $r$-sparse. In particular, $D_r = \{(\mathbf{x}, \mathbf{x}') \text{ s.t. } \|\mathbf{x}\|_0, \|\mathbf{x}'\|_0, \|\mathbf{x}' - \mathbf{x}\|_0 \leq r\}$.*

Generalized orthogonal matching pursuit (OMP) [4, 43, 44] is a greedy algorithm that keeps track of a support set $\mathcal{S}$ of non-zero coefficients, and expands the support based on the largest gradient magnitudes, applied to the marginal liklihood optimization problem, $\mathcal{S}_{i+1} = \mathcal{S}_i \cup \arg\max_{j \notin \mathcal{S}} |\nabla_{\boldsymbol{\rho}}\mathcal{L}(\boldsymbol{\rho})|_j$. Algorithm 1 generalizes OMP [62] by allowing more general support expansion schedules $\mathcal{K}$, and specializes the support expansion criterion using the special problem structure exposed by Lemma 1.

## 5.2 The Convex Parameterization

The NMLL $\mathcal{L}$ of a GP (1) is the sum of a convex function $(\cdot)^{-1}$ and a concave function $\log\det(\cdot)$ of $\mathbf{K}$, and is therefore not generally convex as a function of the hyper-parameters $\boldsymbol{\theta}$, including the robust variances $\boldsymbol{\rho}$. Here, we propose a re-parameterization that allows us to prove strong convexity guarantees of the associated NMLL. In particular, we let $\boldsymbol{\rho}(\mathbf{s}) = \text{diag}(\mathbf{K}_0) \odot ((1 - \mathbf{s})^{-1} - 1)$, where $\mathbf{K}_0 := k(\mathbf{X}, \mathbf{X}) + \sigma^2\mathbf{I}$ and the inverse is element-wise. Note that $\boldsymbol{\rho}(\mathbf{s})$ is a diffeomorphism that maps $\mathbf{s}$ from the compact domain $\mathbf{s} \in [0, 1]^n$ to the entire range of $\boldsymbol{\rho} \in [0, \infty]^n$.

Henceforth, we refer to the original $\boldsymbol{\rho}$ as the *canonical* or $\boldsymbol{\rho}$-parameterization and the newly proposed $\boldsymbol{\rho}(\mathbf{s})$ as the *convex* or $\mathbf{s}$-parameterization. Lemma 4 shows the Hessian of the $\mathbf{s}$-parameterization.

**Lemma 4.** *[Reparameterized Hessian] Let $\mathbf{K_s} = k(\mathbf{X}, \mathbf{X}) + \sigma^2\mathbf{I} + \mathbf{D}_{\boldsymbol{\rho}(\mathbf{s})}$, $\hat{\mathbf{K}}_\mathbf{s} = \text{diag}(\mathbf{K_s})^{-1/2}\mathbf{K_s}\,\text{diag}(\mathbf{K_s})^{-1/2}$, and $\hat{\boldsymbol{\alpha}} = \hat{\mathbf{K}}_\mathbf{s}^{-1}\text{diag}(\mathbf{K_s})^{-1/2}\mathbf{y}$. Then*

$$\mathbf{H_s}[-2\mathcal{L}(\boldsymbol{\rho}(\mathbf{s})] = \mathbf{D}_{1-\mathbf{s}}^{-1}\left[2\left(\hat{\boldsymbol{\alpha}}\hat{\boldsymbol{\alpha}}^\top \odot (\hat{\mathbf{K}}^{-1} - \mathbf{I})\right) + 2\,\text{diag}(\hat{\mathbf{K}}^{-1}) - (\hat{\mathbf{K}}^{-1} \odot \hat{\mathbf{K}}^{-1})\right]\mathbf{D}_{1-\mathbf{s}}^{-1}.$$

Based on this representation, we now derive conditions on the eigenvalues of $\hat{\mathbf{K}}$ that imply the $m$-strong convexity and $M$-smoothness of the NMLL.

**Lemma 5.** *[Strong Convexity via Eigenvalue Condition] Let $\hat{\mathbf{K}}_\mathbf{s}$ as in Lemma 4. Then $\mathbf{H_s} \succ m$ if*

$$\lambda_{\min}\hat{\lambda}_{\min}^2\frac{(2\hat{\lambda}_{\max}^{-1} - \hat{\lambda}_{\min}^{-2} - m)}{2(1 - \lambda_{\min}/\lambda_{\max})} > \|\mathbf{y}\|_2^2, \tag{4}$$

*where $\lambda_{\min,\max}$ (resp. $\hat{\lambda}_{\min,\max}$) are the smallest and largest eigenvalues of $\mathbf{K_s}$, respectively $\hat{\mathbf{K}}_\mathbf{s}$.*

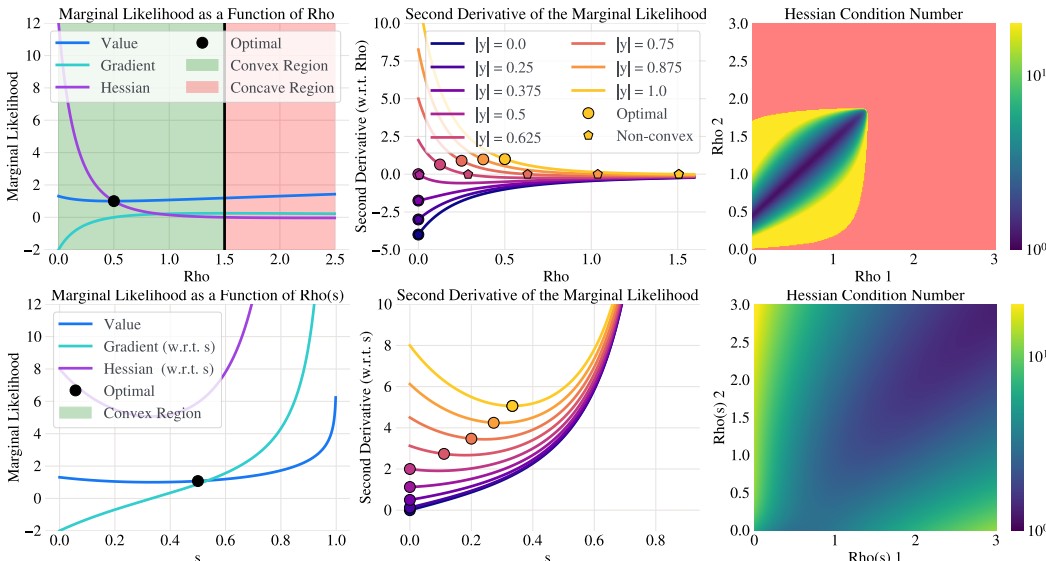

Figure 3: *Top:* The behavior of the $-\log \mathcal{L}(\rho)$ with respect to the canonical parameterization of $\boldsymbol{\rho}$. *Bottom:* The behavior of $-\log \mathcal{L}(\rho(\mathbf{s}))$, highlighting the convexity property. *Left:* The value, and first two derivatives of $-\log \mathcal{L}$ for a 1d example. *Center:* The second derivatives of a 1d $-\log \mathcal{L}$ as a function of $|y|$. The $\mathbf{s}$-parameterization is everwhere convex for all considered $|y|$, while the canonical $\boldsymbol{\rho}$-parameterization is only convex around the origin and only for $|y| > 0.5$. *Right:* The heatmaps highlight that the original parameterization is non-convex (red) for larger values of $\rho$, and quickly becomes ill-conditioned, whereas the parameterization $\boldsymbol{\rho}(\mathbf{s})$ is convex and much better conditioned.

The behavior Lemma 5 predicts is surprising and validated in Fig. 3. Notably, the denominator "blows up" as $\mathbf{K}$ becomes close to unitary, making the inequality more likely to be satisfied, an indication that the convexity property of the NMLL is intimately linked to the RIP (Def. 2). Note that Lemma 5 is a condition for non-support-restricted convexity, which is stronger than is necessary for the approximation guarantees that rely on restricted convexity (Def. 3). However, sparse eigenvalues are generally difficult to compute exactly. Fortunately, covariance matrices of GPs naturally tend to exhibit a property that facilitates a different sufficient condition for convexity for all $\mathbf{s} \in [0,1]^n$.

**Definition 6** (Diagonal Dominance). *A matrix $\mathbf{A}$ is said to be $\delta$-diagonally dominant if the elements $a_{ij}$ satisfy $\sum_{i \neq j} |a_{ij}| < \delta |a_{ii}|$ for all $i$.*

Intuitively, the $\rho_i(s)$ that are selected to be non-zero by the greedy algorithm take on large values, further encouraging the diagonal dominance of the sub-matrix of $\mathbf{K}$ associated with the support of $\boldsymbol{\rho}$. For this reason, the following condition on $\mathbf{K}_0$ is sufficient to guarantee convexity for all $\mathbf{s} \in [0,1]^n$.

**Lemma 7.** *[Strong Convexity via Diagonal Dominance] Let $m > 0$ and $\mathbf{K}_0$ be $\delta$-diagonally dominant with $\delta < \left((5-m) - \sqrt{25 - 9m + 17}\right)/4 \leq (5 - \sqrt{17})/4 \approx 0.44$ and*

$$\lambda_{\min}(\mathbf{K}_0)(1-\delta)^2 \frac{2(1+\delta)^{-1} - (1-\delta)^{-2} - m}{2(1 - (1-\delta)/(1+\delta))} \geq \|\mathbf{y}\|_2^2.$$

*Then the NMLL is $m$-strongly convex for all $\mathbf{s} \in [0,1]^n$, i.e. $\boldsymbol{\rho}(\mathbf{s}) \in [0,\infty]^n$.*

We attain similar results for $M$-smoothness, see Lemma 13 and Lemma 14 in the Appendix. Having proven $m$-convexity and $M$-smoothness conditions, we appeal to the results of Elenberg et al. [23].

**Theorem 8.** *[Approximation Guarantee] Let $\mathbf{K}_0 = k(\mathbf{X}, \mathbf{X}) + \sigma^2 \mathbf{I}$ be $\delta$-diagonally dominant, $s_{\max} > 0$ be an upper bound on $\|\mathbf{s}\|_\infty$, and suppose $\|\mathbf{y}\|, \delta$ satisfy the bounds of Lemmas 7 and 14, guaranteeing $m$-convexity and $M$-smoothness of the NMLL for some $m > 0$, $M > 1/(1 - s_{\max})^2$. Let $\mathbf{s}_{\mathrm{OMP}}(r)$ be the $r$-sparse vector attained by OMP on the NMLL objective for $r$ steps, and let $\mathbf{s}_{\mathrm{OPT}}(r) = \arg\max_{\|\mathbf{s}\|_0 = r, \|\mathbf{s}\|_\infty \leq s_{\max}} \mathcal{L}(\boldsymbol{\rho}(\mathbf{s}))$ be the optimal $r$-sparse vector. Then for any $2r \leq n$,*

$$\tilde{\mathcal{L}}\left(\boldsymbol{\rho}(\mathbf{s}_{\mathrm{OMP}}(r))\right) \geq \left(1 - e^{-m/M}\right) \tilde{\mathcal{L}}\left(\boldsymbol{\rho}(\mathbf{s}_{\mathrm{OPT}}(r))\right),$$

*where $\tilde{\mathcal{L}}(\cdot) = \mathcal{L}(\cdot) - \mathcal{L}(\mathbf{0})$ is normalized so that $\max_{\mathbf{s}_{\mathcal{S}}} \tilde{\mathcal{L}}(\mathbf{s}_{\mathcal{S}}) \geq 0$ for any support $\mathcal{S}$.*

A limitation of the theory is that it assumes the other hyper-parameters of the GP model to be constant, as doing otherwise would introduce the non-convexity that is common to most marginal likelihood optimization problems. In practice, we typically optimize $\rho$ jointly with the other hyper-parameters of the model in each iteration of RRP, as this yields improved performance, see App. D.5 for details.

# 6  Empirical Results

We evaluate the empirical performance of RRP against various baselines on a number of regression and Bayesian Optimization problems. Specifically, we compare against a standard GP with a Matern-5/2 kernel ("Standard GP"), data pre-processing through Ax's adaptive winsorization procedure ("Adapt. Wins.") [12], and a power transformation ("Power Transf.") [15]. Further, we also consider a Student-$t$ likelihood model from Jylänki et al. [32] ("Student-$t$"), the trimmed marginal likelihood model from Andrade and Takeda [10] ("Trimmed MLL"), and the RCGP model from Altamirano et al. [2]. Unless stated otherwise, all models are implemented in GPyTorch [25] and all experiments in this section use 32 replications. See Appendix D for additional details.

## 6.1  Regression Problems

**Synthetic**  We first consider the popular Friedman10 and Hartmann6 [21] test functions from the literature. We use two data generating processes: uniform noise, extreme outliers at some fixed value, and heavy-tailed (Student-$t$) noise at true function values. In these experiments, we compare the performance predictive log-likelihood. The results are shown in Fig. 4.

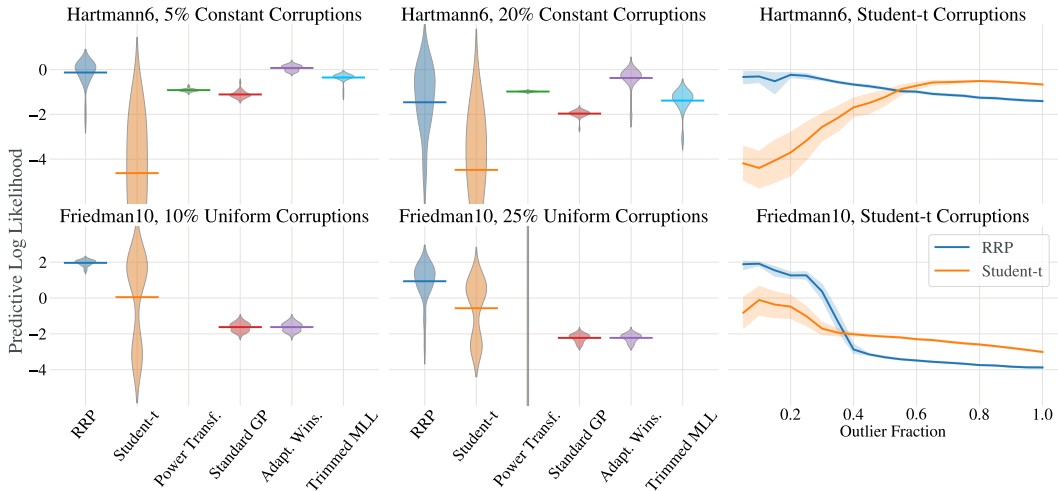

Figure 4:  *Left:* Distribution of predictive test-set log likelihood for various methods. Methods ommitted are those that performed substantially worse. *Right:* Predictive log likelihood as a function of the corruption probability for Student-$t$-distributed corruptions with two degrees of freedom. The GP model with the Student-$t$ likelihood only starts outperforming RRP as the corruption probability increases beyond 40%, and exhibits a large variance in outcomes, which shrinks as the proportion of corruptions increases. All methods not shown were inferior to either RRP or Student-$t$.

**Twitter Flash Crash**  In Fig. 5, we report a comparison to Altamirano et al. [2]'s RCGP on data from the Dow Jones Industrial Average (DJIA) index on April 22-23 2013, which includes a sharp drop at 13:10 on the 23rd. The top panels shows that RCGP exhibits higher robustness than the standard GP, but is still affected by the outliers, when trained on data from the 23rd. RRP is virtually unaffected. Notably, RCGP relies on an a-priori weighting of data points based on the target values' proximity to their median, which can be counter-productive when the outliers are not a-priori separated in the range. To highlight this, we included the previous trading day into the training data for the bottom panels, leading RCGP to assign the *highest* weight to the outlying data points due to their proximity to the target values' median, thereby leading RCGP to "trust" the outliers more than any inlier,

resulting in it being less robust than a standard GP in this scenario. See Appendix D.6 for additional comparisons to RCGP, on data sets from the UCI machine learning repository [34].

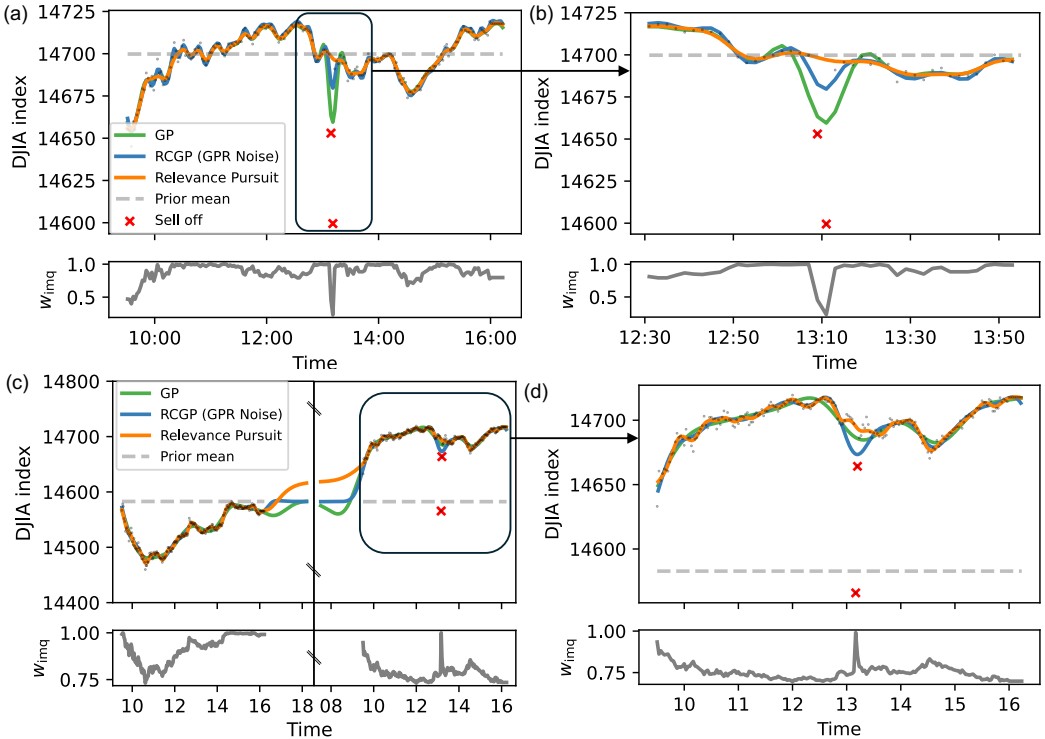

Figure 5: Results on the intra-day data from the Dow Jones Industrial Average (DJIA) index on April 22-23 2013, which includes a sharp drop at 13:10 on the 23rd, see (b) for a detailed view. The accompanying panels labeled $w_{\mathrm{imq}}$ show the function that Altamirano et al. [2]'s RCGP uses to down-weight data points. *Top*: RCGP, exhibits higher robustness than the standard GP, but is still affected by the outliers. The RRP model is virtually unaffected. *Bottom*: Including the previous trading day into the training data in (c), leads RCGP to assign the *highest* weight $w_{\mathrm{imq}}$ to the outlying data points due to their proximity to the target values' median, thereby leading RCGP to be even more affected than a standard GP, see (d) for a detailed view of the results on the data of April 23.

## 6.2 Robust Bayesian Optimization

GPs are commonly used for Bayesian optimization (BO), which is a popular approach to sample-efficient black-box optimization [24]. However, many of the GP models used for BO are sensitive to outliers and may not perform well in settings where such outliers occur. While Martinez-Cantin et al. [46] consider the use of a Student-$t$ likelihood for BO with outliers, the use of other robust GP models has not been thoroughly studied in the literature.

**Experimental setup**  We use Ament et al. [7]'s `qLogNoisyExpectedImprovement` (qLogNEI), a variant of the LogEI family of acquisition functions, 32 replications, and initialize all methods with the same quasi-random Sobol batch for each replication. We follow Hvarfner et al. [31] and plot the true value of the best in-sample point according to the GP model posterior at each iteration. We also include Sobol and an "Oracle", which is a Standard GP that always observes the uncorrupted value, and consider the backward canonical version of relevance pursuit, denoted by RRP, for these experiments. The plots show the mean performance with a bootstrapped 90% confidence interval.

**Synthetic problems**  We consider the popular 6-dimensional Hartmann test function with three different corruption settings: (1) a constant value of 100, (2) a $U[-3, 3]$ distributed value, (3) the objective value for a randomly chosen point in the domain. The results for a 10% corruption probability are shown in Fig. 6. We also include results for a 20% corruption probability in Appendix D.3.

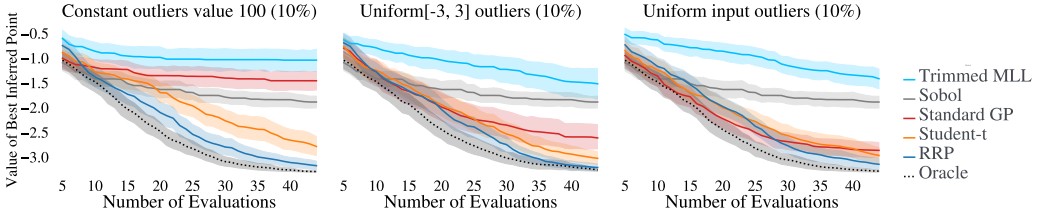

Figure 6: BO results for Hartmann6: *Left:* Relevance pursuit performs well in the case of constant outliers of value 100, almost as well as the oracle. *Middle:* Relevance pursuit performs the best followed by the Student-$t$ likelihood in the case of $U[-3, 3]$. *Right:* Similar to the middle plot, this setting hides the corruptions within the range of the function, making it a challenging task.

**Real-world problems**  We include three real-world problems: A 3D SVM problem, a 5D CNN problem, and a 20D rover trajectory planning problem, see the App. D.2 for details. For SVM and CNN, we simulate random corruptions corresponding to an I/O error, which causes the corresponding ML model to be trained using only a small subset of the training data. For the rover planning problem we follow the setup in [47] with the main difference that we consider a 20D trajectory, and the corruptions are generated randomly, causing the rover to break down at an arbitrary point along its trajectory. In most cases, this results in a smaller reward than the reward of the full trajectory.

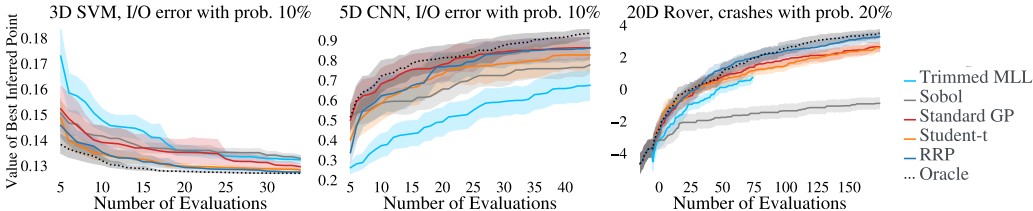

Figure 7: BO results for three real-world problems: *Left:* RRP is competitive with the oracle on the 3D SVM problem. *Middle:* The power transform performs best on the 5D CNN problem, outperforming RRP as well as the Oracle. *Right:* RRP performs well on the 20D Rover problem.

## 7  Conclusion and Future Work

**Contributions**  Robust Gaussian Processes via Relevance Pursuit (RRP) provides a novel and principled way to perform robust GP regression. It permits efficient and robust inference, performs well across a variety of label corruption settings, retains good performance in the absence of corruptions, and is flexible, e.g., can be used with any mean or kernel function. Our method can be readily applied to both robust regression problems as well as applications such as Bayesian optimization and is available through `BoTorch` [13]. Importantly, it also provides theoretical approximation guarantees.

**Limitations**  As our approach does not explicitly consider the locations of the data points in the outlier identification, it may be outperformed by other methods if the underlying noise is heteroskedastic and location-dependent. On the other hand, those methods generally do not perform well in the presence of sparse, location-independent data corruptions.

**Extensions**  Promising extensions of this work include performing Bayesian model averaging, i.e., average the predictions of the different possible sparsity models according to their likelihoods instead of using a MAP estimate, applying RRP to specialized models such as Lin et al. [41]'s scalable learning-curve model for AutoML applications, and Ament et al. [8]'s model for sustainable concrete. On a higher level, the approach of combining greedy optimization algorithms with Bayesian model selection and leveraging a convex parameterization to achieve approximation guarantees might apply to other parameters that are optimized using the MLL objective: length-scales of stationary kernels, coefficients of additive kernels, inducing inputs, and even related model classes like Tipping [60]'s Sparse Bayesian Learning (SBL), which seeks to identify sparse linear models and is intimately linked to greedy matching pursuits [9]. Overall, the approach has the potential to lead to theoretical guarantees, new insights, and performance improvements to widely-adopted Bayesian models.

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

# A    Additional Details on the Model

Algorithm 2 below is the "backward" variant of Algorithm 1 from Sec 4. As its name suggests, the main difference compared to the "forward" variant is that rather than building up a set of "outliers", it starts from a (typically large) set of "outliers" and iteratively *removes* those data points from the set that have the smallest inferred data-point-dependent noise variance $\rho_i$.

While we have not derived theoretical guarantees for this "backward" version, we have found it to generally behave similarly to the "forward" version in terms of performance and robustness. One empirical observation from our studies is that while the "forward" version tends to perform slightly better than the "backward" version if there are only few outliers, the opposite is true if the outlier frequency is very high. This behavior is rather intuitive and illustrates that relevance pursuit is particularly well-suited to identify sparse, low-cardinality subsets (note that in the "backward" variant under large corruptions, the uncorrupted data points can be viewed as the sparse subset that needs to be identified).

---

**Algorithm 2** Relevance Pursuit (Backward Algorithm)

---

**Require:** $\mathbf{X}$, $\mathbf{y}$, schedule $\mathcal{K} = (k_1, k_2, \dots)$
    Initialize $\mathcal{S}_0^c \subseteq \{1, \dots, n\}$ (typically $\mathcal{S}_0^c = \{1, \dots, n\}$)
    **for** $k_i$ in $\mathcal{K}$ **do**
        Optimize ML: $\boldsymbol{\rho}_{\mathcal{S}_i^c}^* \leftarrow \arg\max_{\boldsymbol{\rho}_{\mathcal{S}_i^c}} \mathcal{L}(\boldsymbol{\rho}_{\mathcal{S}_i^c})$,   where $\boldsymbol{\rho}_{\mathcal{S}_i^c} = \{\boldsymbol{\rho} : \rho_j = 0, \ \forall j \notin \mathcal{S}_i^c\}$
        Compute the set $\mathcal{R}_i$ containing the $k_i$ elements of $\mathcal{S}_i^c$ with smallest inferred variance:
            $\mathcal{R}_i \leftarrow \{j_i^1, \dots, j_i^{k_i}\}$ where $j_i^l \in \mathcal{S}_i^c$ such that $\rho_{\mathcal{S}_i^c}^*(j_i^l) \leq \rho_{\mathcal{S}_i^c}^*(j_i^{l'})$ for $l < l'$
        $\mathcal{S}_{i+1}^c \leftarrow \mathcal{S}_i^c \setminus \mathcal{R}_i$
    $\mathcal{S}_i \leftarrow \{1, \dots, n\} \setminus \mathcal{S}_i^c$ for each $k_i$
    Compute the marginal likelihood $p(\mathcal{S}_i | \mathbf{X}, \mathbf{y}) \approx p(\mathcal{S}_i, \boldsymbol{\rho}_{\mathcal{S}_i}^*, |\mathbf{X}, \mathbf{y})$
    $\mathcal{S}^* \leftarrow \arg\max_{\mathcal{S}_i} p(\mathcal{S}_i | \mathbf{X}, \mathbf{y}) p(\mathcal{S}_i)$.
    Return $\mathcal{S}^*$, $\boldsymbol{\rho}_{\mathcal{S}^*}^*$.

---

# B    Additional Background on the Theory

## B.1    Submodular Functions

Krause and Golovin [36] provides a survey on the maximization of general submodular functions. Here, we focus on applications of submodularity to sparse regression and Gaussian process models.

**Sparse Regression**    Das and Kempe [20] showed that the subset selection problem of regression features with an $R^2$ objective satisfies a weak submodularity property, which can be invoked to prove approximation guarantees for the greedy maximization of the objective. Elenberg et al. [23] generalized this work by proving that any log likelihood function exhibiting restricted strong concavity gives rise to the weak submodularity of the associated subset selection problem, which can be invoked to prove approximation guarantees for the greedy algorithm. Karaca et al. [33] contains a guarantee for the backward algorithm applied to the maximization of submodular set functions.

**Gaussian Processes**    Submodularity has also found application to Gaussian process models. For a sensor placement problem, Krause et al. [38] proved that the mutual information (MI) criterion, capturing the reduction in uncertainty in the entire search space, can be a submodular function. In this case, MI is not monotonic everywhere, but monotonic for small sets ($2k$) of sensors, which is sufficient to apply Nemhauser's guarantee for sparse sets of sensors up to size $k$ [50]. Relatedly, the "myopic" joint entropy of a set of observables is unconditionally submodular as a consequence of the "information never hurts" principle [37], but generally leads to lower-quality sensor placements than the MI criterion. Srinivas et al. [57] used the submodularity of the joint entropy in order to prove regret bounds for the convergence of a GP-based BO algorithm using the upper-confidence bound acquisition function.

Elenberg et al. [23] proved that any log likelihood function exhibiting restricted strong concavity and smoothness implies the weak submodularity of the associated subset selection problem.

**Definition 9** (Submodularity Ratios [23]). *Let $\mathcal{A}, \mathcal{B} \subset [n]$ be two disjoint sets, and $f : 2^{[n]} \to \mathbb{R}$. The submodularity ratio of $\mathcal{B}$ with respect to $\mathcal{A}$ is defined by*

$$\gamma_{\mathcal{B},\mathcal{A}} = \sum_{i \in \mathcal{A}} \left( f(\mathcal{B} \cup \{i\}) - f(\mathcal{B}) \right) / \left( f(\mathcal{B} \cup \mathcal{A}) - f(\mathcal{B}) \right)$$

*The submodularity ratio of a set $\mathcal{C}$ with respect to an integer $k$ is defined by*

$$\gamma_{\mathcal{C},k} = \min_{\mathcal{B},\mathcal{A}} \gamma_{\mathcal{B},\mathcal{A}} \qquad \text{such that} \qquad \mathcal{A} \cap \mathcal{B} = \emptyset, \qquad \mathcal{B} \subseteq \mathcal{C}, \qquad \text{and} \qquad |\mathcal{A}| \leq k.$$

*Then given $\gamma > 0$, a function is $\gamma$-weakly submodular at a set $\mathcal{C}$ with respect to $k$ if $\gamma_{\mathcal{C},k} \geq \gamma$.*

**Theorem 10** (Weak Submodularity via RSC [23]). *The submodularity ratio $\gamma_{\mathcal{S},k}$ can be bound below using the restricted convexity and smoothness parameters $m_{|\mathcal{S}|+k}$ and $M_{|\mathcal{S}|+k}$,*

$$\gamma_{\mathcal{S},k} \geq m_{|\mathcal{S}|+k} \big/ M_{|\mathcal{S}|+k}.$$

**Theorem 11** (OMP Approximation Guarantee [23]). *Let $\mathbf{x}_{\mathrm{OMP}}(r)$ be the $r$-sparse vector selected by OMP, and $\mathbf{x}_{\mathrm{OPT}}(r) = \arg\min_{\|\mathbf{x}\|_0 = r} f(\mathbf{x})$ be the optimal $r$-sparse vector. Then*

$$f\left(\mathbf{x}_{\mathrm{OMP}}(r)\right) \geq \left( 1 - e^{-m_{2r}/M_{2r}} \right) f\left(\mathbf{x}_{\mathrm{OPT}}(r)\right),$$

*where $m_{2r}$, $M_{2r}$ are the restricted strong convexity and smoothness parameters of $f$, respectively.*

## C  Theoretical Results and Proofs

**Lemma 1.** *[Optimal Robust Variances] Let $\mathcal{D}_{\backslash i} = \{(\mathbf{x}_j, y_j) : j \neq i\}$, $\boldsymbol{\rho} = \boldsymbol{\rho}_{\backslash i} + \rho_i \mathbf{e}_i$, where $\boldsymbol{\rho}, \boldsymbol{\rho}_{\backslash i} \in \mathbb{R}_+^n$, $[\boldsymbol{\rho}_{\backslash i}]_i = 0$, and $\mathbf{e}_i$ is the $i$th canonical basis vector. Then keeping $\boldsymbol{\rho}_{\backslash i}$ fixed,*

$$\rho_i^* = \arg\max_{\rho_i} \mathcal{L}\left(\boldsymbol{\rho}_{\backslash i} + \rho_i \mathbf{e}_i\right) = \left[ (y_i - \mathbb{E}[y(\mathbf{x}_i)|\mathcal{D}_{\backslash i}])^2 - \mathbb{V}[y(\mathbf{x}_i)|\mathcal{D}_{\backslash i}] \right]_+, \tag{3}$$

*where $y(\mathbf{x}_i) = f(\mathbf{x}_i) + \epsilon_i$. These quantities can be expressed as functions of $\boldsymbol{\Sigma}^{-1} = (\mathbf{K} + \mathbf{D}_{\sigma^2 + \boldsymbol{\rho}})^{-1}$:*

$$\mathbb{E}[y(\mathbf{x}_i)|\mathcal{D}_{\backslash i}]^2 = y_i - \left[ \boldsymbol{\Sigma}^{-1} \mathbf{y} \right]_i \big/ \left[ \boldsymbol{\Sigma}^{-1} \right]_{ii}, \qquad \text{and} \qquad \mathbb{V}[y(\mathbf{x}_i)|\mathcal{D}_{\backslash i}] = 1 \big/ \left[ \boldsymbol{\Sigma}^{-1} \right]_{ii},$$

*where $\mathbf{D}_{\sigma^2 + \boldsymbol{\rho}}$ is a diagonal matrix whose entries are $\sigma^2 + \boldsymbol{\rho}$.*

*Proof.* First, we partition the covariance matrix $\mathbf{K} + \sigma^2 \mathbf{I} + \mathbf{D}_{\boldsymbol{\rho}}$ to separate the effect of $\rho_i$ and use the block matrix inverse

$$\boldsymbol{\Sigma}^{-1} = (\mathbf{K} + \mathbf{D}_{\boldsymbol{\rho} + \sigma^2})^{-1} = \begin{bmatrix} \boldsymbol{\Sigma}_{\backslash i}^{-1} + \mathbf{u}\beta_i \mathbf{u}^\top & -\mathbf{u}\beta_i \\ -\mathbf{u}^\top \beta_i & \beta_i \end{bmatrix}, \tag{5}$$

where

$$\begin{aligned} \boldsymbol{\Sigma}_{\backslash i} &= k(\mathbf{X}_{\backslash i}, \mathbf{X}_{\backslash i}) + \mathbf{D}_{\boldsymbol{\rho}_{\backslash i} + \sigma^2}, \\ \mathbf{u} &= \boldsymbol{\Sigma}_{\backslash i}^{-1} k(\mathbf{X}_{\backslash i}, \mathbf{x}_i), \qquad \text{and} \\ \beta_i &= \left( [k(\mathbf{x}_i, \mathbf{x}_i) + \sigma^2 + \rho_i] - k(\mathbf{x}_i, \mathbf{X}_{\backslash i}) \boldsymbol{\Sigma}_{\backslash i}^{-1} k(\mathbf{X}_{\backslash i}, \mathbf{x}_i) \right)^{-1}. \end{aligned} \tag{6}$$

**Quadratic Term**  With the expression for the inverse of $\boldsymbol{\Sigma}$ above, we can write the quadratic term of the log likelihood as

$$\begin{aligned} \mathbf{y}^\top (\mathbf{K} + \mathbf{D}_{\boldsymbol{\rho} + \sigma^2})^{-1} \mathbf{y} &= \mathbf{y}_{\backslash i}^\top (\boldsymbol{\Sigma}_{\backslash i}^{-1} + \mathbf{u}\beta_i \mathbf{u}^\top) \mathbf{y}_{\backslash i} - 2 \mathbf{y}_{\backslash i}^\top \mathbf{u}\beta_i y_i + y_i^2 \beta_i \\ &= \mathbf{y}_{\backslash i}^\top \boldsymbol{\Sigma}_{\backslash i}^{-1} \mathbf{y}_{\backslash i} + \beta_i (\mathbf{y}_{\backslash i}^\top \mathbf{u} - y_i)^2. \end{aligned} \tag{7}$$

**Determinant Term**  The determinant of a block matrix is given by

$$\left| \begin{bmatrix} \mathbf{A} & \mathbf{B} \\ \mathbf{C} & \mathbf{D} \end{bmatrix} \right| = |\mathbf{A}| |\mathbf{D} - \mathbf{C}\mathbf{A}^{-1}\mathbf{B}|. \tag{8}$$

Applying this identity to $\boldsymbol{\Sigma} = (\mathbf{K} + \mathbf{D}_{\boldsymbol{\rho} + \sigma^2})$, we get

$$|\mathbf{K} + \mathbf{D}_{\boldsymbol{\rho} + \sigma^2}| = |\boldsymbol{\Sigma}_{\backslash i}| \beta_i^{-1}. \tag{9}$$

**Log Marginal Likelihood**

$$2(\mathcal{L}(\boldsymbol{\rho}) - \mathcal{L}(\boldsymbol{\rho}_{\setminus i})) = -\beta_i(\mathbf{y}_{\setminus i}^\top \mathbf{u} - y_i)^2 - \log(\beta_i^{-1}). \tag{10}$$

Noting that $\partial_{\rho_i}\beta_i = -\beta_i^2$, the derivative of the difference in log marginal likelihood w.r.t. $\rho_i$, is

$$\partial_{\rho_i} 2(\mathcal{L}(\boldsymbol{\rho}) - \mathcal{L}(\boldsymbol{\rho}_{\setminus i})) = (\mathbf{y}_{\setminus i}^\top \mathbf{u} - y_i)^2 \beta_i^2 - \beta_i. \tag{11}$$

While $\beta_i = 0$ is a root of the derivative, we ignore this solution since $\beta_i$ is never zero when $\sigma^2 > 0$. Therefore, the remaining stationary point is $\beta_i^{-1} = (\mathbf{y}_{\setminus i}^\top \mathbf{u} - y_i)^2$. Since we constrain $\rho \geq 0$, this point might not always be attainable. However, because there is only a single stationary point with respect to $\beta_i$ when $\sigma^2 > 0$, and $\beta_i$ is a strictly decreasing function of $\rho_i$, it follows that the marginal likelihood is monotonic as a function of $\rho_i$ to both the left and the right of the stationary point. Therefore, the optimal constraint $\rho_i$ is simply the optimal unconstrained value, projected into the feasible space. In particular, solving $\beta_i^{-1} = (\mathbf{y}_{\setminus i}^\top \mathbf{u} - y_i)^2$ for $\rho_i$ and projecting to the non-negative half-line, we get

$$\rho_i = \left[ \left(\mathbf{y}_{\setminus i}^\top \mathbf{u} - y_i\right)^2 - \left(k(\mathbf{x}_i, \mathbf{x}_i) + \sigma^2 - k(\mathbf{x}_i, \mathbf{X}_{\setminus i})\boldsymbol{\Sigma}_{\setminus i}^{-1}k(\mathbf{X}_{\setminus i}, \mathbf{x}_i)\right) \right]_+. \tag{12}$$

Lastly, note

$$\mathbf{y}_{\setminus i}^\top \mathbf{u} - y_i = \mathbf{y}_{\setminus i}^\top(k(\mathbf{X}_{\setminus i}, \mathbf{X}_{\setminus i}) + \mathbf{D}_{\sigma^2 + \boldsymbol{\rho}_{\setminus i}})^{-1}k(\mathbf{X}_{\setminus i}, \mathbf{x}_i) - y_i = \mathbb{E}[y(\mathbf{x}_i)|\mathcal{D}_{\setminus i}] - y_i,$$

$$k(\mathbf{x}_i, \mathbf{x}_i) - k(\mathbf{x}_i, \mathbf{X}_{\setminus i})\boldsymbol{\Sigma}_{\setminus i}^{-1}k(\mathbf{X}_{\setminus i}, \mathbf{x}_i) + \sigma^2 = \mathbb{V}[y(\mathbf{x}_i)|\mathcal{D}_{\setminus i}]. \tag{13}$$

As stated by Rasmussen et al. [54] (P. 117, Eq. 5.12), originally shown by Sundararajan and Keerthi [58], these quantities can be expressed as simple functions of $\boldsymbol{\Sigma}^{-1}$:

$$\mathbb{E}[y(\mathbf{x}_i)|\mathcal{D}_{\setminus i}]^2 = y_i - \left[\boldsymbol{\Sigma}^{-1}\mathbf{y}\right]_i / \left[\boldsymbol{\Sigma}^{-1}\right]_{ii}$$

$$\mathbb{V}[y(\mathbf{x}_i)|\mathcal{D}_{\setminus i}] = 1 / \left[\boldsymbol{\Sigma}^{-1}\right]_{ii}. \tag{14}$$

Therefore, all LOO predictive values can be computed in $\mathcal{O}(n^3)$ or faster, if an inducing point method is used for $\mathbf{K}$. $\qquad\square$

The following is a preliminary result for our analysis of the log marginal likelihood w.r.t. $\boldsymbol{\rho}$.

**Lemma 12.** *The gradient and Hessian of the log marginal likelihood $\mathcal{L}$ with respect to $\boldsymbol{\rho}$ are given by*

$$-2\nabla_{\boldsymbol{\rho}}[\mathcal{L}] = \mathrm{diag}(\mathbf{K}^{-1} - \boldsymbol{\alpha}\boldsymbol{\alpha}^\top), \qquad \textit{and} \qquad -2\mathbf{H}_{\boldsymbol{\rho}}[\mathcal{L}] = (2\boldsymbol{\alpha}\boldsymbol{\alpha}^\top - \mathbf{K}^{-1}) \odot \mathbf{K}^{-1},$$

*where $\mathbf{K} = \mathbf{K}_0 + \mathbf{D}_{\boldsymbol{\rho}}$ for some base covariance matrix $\mathbf{K}_0$ and $\boldsymbol{\alpha} = \mathbf{K}^{-1}\mathbf{y}$.*

*Proof.* Let $\boldsymbol{\alpha} = \mathbf{K}^{-1}\mathbf{y}$. Regarding the gradient, note that $\partial_{\rho_i}\mathbf{K} = \mathbf{e}_i\mathbf{e}_i^\top$, where $\mathbf{e}_i$ is the canonical basis vector with a one as the $i$th element, and based on Equation 5.9 of Rasmussen et al. [54],

$$-2\partial_{\rho_i}[\mathcal{L}] = \mathrm{tr}\left((\mathbf{K}^{-1} - \boldsymbol{\alpha}\boldsymbol{\alpha}^\top)\partial_{\rho_i}\mathbf{K}\right)$$

$$= \mathrm{tr}((\mathbf{K}^{-1} - \boldsymbol{\alpha}\boldsymbol{\alpha}^\top)\mathbf{e}_i\mathbf{e}_i^\top)$$

$$= \mathbf{e}_i^\top(\mathbf{K}^{-1} - \boldsymbol{\alpha}\boldsymbol{\alpha}^\top)\mathbf{e}_i$$

$$= (\mathbf{K}^{-1} - \boldsymbol{\alpha}\boldsymbol{\alpha}^\top)_{ii}.$$

Regarding the second derivatives, according to Dong et al. [22],

$$\partial_{\theta_i}\partial_{\theta_j}[\log|\mathbf{K}|] = \mathrm{tr}(\mathbf{K}^{-1}[\partial_{\theta_i}\partial_{\theta_j}\mathbf{K}] - \mathbf{K}^{-1}[\partial_{\theta_i}\mathbf{K}]\mathbf{K}^{-1}[\partial_{\theta_j}\mathbf{K}])$$

$$\partial_{\theta_i}\partial_{\theta_j}[\mathbf{y}^\top\mathbf{K}^{-1}\mathbf{y}] = 2\boldsymbol{\alpha}^T[\partial_{\theta_i}\mathbf{K}]\mathbf{K}^{-1}[\partial_{\theta_j}\mathbf{K}]\boldsymbol{\alpha} - \boldsymbol{\alpha}^\top[\partial_{\theta_i}\partial_{\theta_j}\mathbf{K}]\boldsymbol{\alpha}.$$

Therefore,

$$-2\partial_{\rho_i}\partial_{\rho_j}\mathcal{L} = \mathrm{tr}\left((\mathbf{K}^{-1} - \boldsymbol{\alpha}\boldsymbol{\alpha}^T)[\partial_{\rho_i}\partial_{\rho_j}\mathbf{K}] - (\mathbf{K}^{-1} - 2\boldsymbol{\alpha}\boldsymbol{\alpha}^\top)([\partial_{\rho_i}\mathbf{K}]\mathbf{K}^{-1}[\partial_{\rho_j}\mathbf{K}])\right)$$

$$= \mathrm{tr}\left((2\boldsymbol{\alpha}\boldsymbol{\alpha}^\top - \mathbf{K}^{-1})([\mathbf{K}^{-1}]_{ij}\mathbf{e}_i\mathbf{e}_j^\top)\right)$$

$$= \mathrm{tr}\left(\mathbf{e}_j^\top(2\boldsymbol{\alpha}\boldsymbol{\alpha}^\top - \mathbf{K}^{-1})\mathbf{e}_i\right)[\mathbf{K}^{-1}]_{ij}$$

$$= [2\boldsymbol{\alpha}\boldsymbol{\alpha}^\top - \mathbf{K}^{-1}]_{ij}[\mathbf{K}^{-1}]_{ij}$$

$$= \left[(2\boldsymbol{\alpha}\boldsymbol{\alpha}^\top - \mathbf{K}^{-1}) \odot \mathbf{K}^{-1}\right]_{ij},$$

since $[\partial_{\rho_i}\partial_{\rho_j}\mathbf{K}] = \mathbf{0}$. The third equality is due to the invariance of the trace to circular shifts of its argument. The forth equality is due to the symmetry of the matrix in brackets. $\qquad\square$

## C.1 Strong Convexity and Smoothness of the Reparameterized Robust Marginal Likelihood

Here, we re-parameterize $\rho(\mathbf{s}) = \text{diag}(\mathbf{K}_0) \odot ((1-\mathbf{s})^{-1} - 1)$, and attain strong convexity for all inputs $\mathbf{s}$, if conditions on the eigenvalues of the covariance matrix and the norm of the data vector $\|\mathbf{y}\|$ are met. The convexity result is surprising in two ways: the negative log marginal likelihood of GPs is generally a non-convex function, and in addition, the negative log likelihoods of many alternative robust regression methods like the Student-$t$ likelihood or $\alpha$-stable likelihoods are non-convex, and even Huber's proposal is non-*strongly*-convex.

**Lemma 4.** *[Reparameterized Hessian]* Let $\mathbf{K_s} = k(\mathbf{X}, \mathbf{X}) + \sigma^2 \mathbf{I} + \mathbf{D}_{\rho(\mathbf{s})}$, $\hat{\mathbf{K}}_\mathbf{s} = \text{diag}(\mathbf{K_s})^{-1/2} \mathbf{K_s} \text{diag}(\mathbf{K_s})^{-1/2}$, and $\hat{\alpha} = \hat{\mathbf{K}}_\mathbf{s}^{-1} \text{diag}(\mathbf{K_s})^{-1/2}\mathbf{y}$. *Then*

$$\mathbf{H_s}[-2\mathcal{L}(\rho(\mathbf{s})] = \mathbf{D}_{1-\mathbf{s}}^{-1} \left[ 2\left(\hat{\alpha}\hat{\alpha}^\top \odot (\hat{\mathbf{K}}^{-1} - \mathbf{I})\right) + 2\,\text{diag}(\hat{\mathbf{K}}^{-1}) - (\hat{\mathbf{K}}^{-1} \odot \hat{\mathbf{K}}^{-1}) \right] \mathbf{D}_{1-\mathbf{s}}^{-1}.$$

*Proof.* Using the chain-rule, the Hessian $\mathbf{H_s}[\mathcal{L}]$ can be expressed as a function of the Jacobian $\mathbf{J_s}[\rho(\mathbf{s})] = \mathbf{D}_{\partial\rho(\mathbf{s})}$, which is diagonal since $\rho(\mathbf{s})$ is an element-wise function, and the second derivatives $\partial_s^2 \rho(s_i)$. Then

$$-2\mathbf{H_s}[\mathcal{L}] = -\mathbf{J_s}[\rho]^\top \mathbf{H}_\rho[2\log\mathcal{L}]\mathbf{J_s}[\rho] + \mathbf{D}_{\nabla_\rho[-2\mathcal{L}]}\mathbf{D}_{\partial_s^2\rho}$$
$$= \mathbf{D}_{\rho'(\mathbf{s})}[(2\alpha\alpha^\top - \mathbf{K}^{-1}) \odot \mathbf{K}^{-1}]\mathbf{D}_{\rho'(\mathbf{s})} + \text{diag}(\mathbf{K}^{-1})\mathbf{D}_{\rho''} - \mathbf{D}_{\rho'}(\mathbf{K}^{-1} \circ \mathbf{K}^{-1})\mathbf{D}_{\rho'},$$

where we substituted the relevant expressions from Lemma 12. Further substituting $\rho(\mathbf{s})_i = [\mathbf{K}_0]_{ii}[(1-s_i)^{-1} - 1]$, $\rho'(\mathbf{s})_i = [\mathbf{K}_0]_{ii}(1-s_i)^{-2}$, and $\rho''(\mathbf{s})_i = 2[\mathbf{K}_0]_{ii}(1-s_i)^{-3}$, noting that $\mathbf{K} = \mathbf{K}_0 + \text{diag}(\mathbf{K}_0)[(1-s)^{-1} - 1] = (\mathbf{K}_0 - \text{diag}(\mathbf{K}_0)) + \text{diag}(\mathbf{K}_0)(1-s)^{-1}$, and algebraic manipulation finish the proof. $\square$

**Lemma 5.** *[Strong Convexity via Eigenvalue Condition]* Let $\hat{\mathbf{K}}_\mathbf{s}$ as in Lemma 4. Then $\mathbf{H_s} \succ m$ if

$$\lambda_{\min}\hat{\lambda}_{\min}^2 \frac{(2\hat{\lambda}_{\max}^{-1} - \hat{\lambda}_{\min}^{-2} - m)}{2(1 - \lambda_{\min}/\lambda_{\max})} > \|\mathbf{y}\|_2^2, \tag{4}$$

*where $\lambda_{\min,\max}$ (resp. $\hat{\lambda}_{\min,\max}$) are the smallest and largest eigenvalues of $\mathbf{K_s}$, respectively $\hat{\mathbf{K}}_\mathbf{s}$.*

*Proof.* We seek to lower-bound the smallest eigenvalue of the Hessian matrix, which—for twice-continuously-differentiable problems—is equivalent to lower and upper bounds of the problem's Hessian matrix. Starting with the result of Lemma 4,

$$\mathbf{H_s}[-2\mathcal{L}(\rho(\mathbf{s})] = \mathbf{D}_{1-\mathbf{s}}^{-1} \left[ 2\left(\hat{\alpha}\hat{\alpha}^\top \odot (\hat{\mathbf{K}}^{-1} - \mathbf{I})\right) + 2\,\text{diag}(\hat{\mathbf{K}}^{-1}) - (\hat{\mathbf{K}}^{-1} \odot \hat{\mathbf{K}}^{-1}) \right] \mathbf{D}_{1-\mathbf{s}}^{-1}$$
$$\succeq 2\left(\hat{\alpha}\hat{\alpha}^\top \odot (\hat{\mathbf{K}}^{-1} - \mathbf{I})\right) + 2\,\text{diag}(\hat{\mathbf{K}}^{-1}) - (\hat{\mathbf{K}}^{-1} \odot \hat{\mathbf{K}}^{-1}).$$

Now, we bound each of the three additive terms independently from below.

**Term 1:**
$$2\,\text{diag}(\hat{\mathbf{K}}^{-1}) \succeq 2\lambda_{\min}(\hat{\mathbf{K}}^{-1})\mathbf{I} = 2\lambda_{\max}(\hat{\mathbf{K}})^{-1}\mathbf{I}.$$

The first inequality comes from $\mathbf{K}$ being positive definite, and the absolute value of the diagonal of a matrix, which is already positive for positive definite matrices, being lower bounded by the minimum eigenvalue of the matrix. The last steps is a basic consequence of the eigenvalues of inverses matrices. Note that the eigenvalues of $\hat{\mathbf{K}}$ can be further bound by the eigenvalues of the original matrix $\mathbf{K}$:

$$\lambda_{\min}(\hat{\mathbf{K}}) = \lambda_{\min}(\text{diag}(\mathbf{K})^{-1/2}\mathbf{K}\,\text{diag}(\mathbf{K})^{-1/2})$$
$$\geq \lambda_{\min}(\text{diag}(\mathbf{K})^{-1/2})^2\lambda_{\min}(\mathbf{K})$$
$$= \lambda_{\min}(\text{diag}(\mathbf{K})^{-1})\lambda_{\min}(\mathbf{K})$$
$$\geq \lambda_{\min}(\mathbf{K}^{-1})\lambda_{\min}(\mathbf{K})$$
$$= \lambda_{\min}(\mathbf{K})/\lambda_{\max}(\mathbf{K}).$$

In a similar way, we can show that $\lambda_{\max}(\hat{\mathbf{K}}) \leq \lambda_{\max}(\mathbf{K})/\lambda_{\min}(\mathbf{K})$, which implies $\lambda_{\max}(\hat{\mathbf{K}})^{-1} \geq \lambda_{\min}(\mathbf{K})/\lambda_{\max}(\mathbf{K})$.

**Term 2:** Next, we have

$$-(\hat{\mathbf{K}}^{-1} \odot \hat{\mathbf{K}}^{-1}) \succeq -\lambda_{\max}(\hat{\mathbf{K}}^{-1})^2 \mathbf{I} = -\lambda_{\min}(\hat{\mathbf{K}})^{-2}\mathbf{I},$$

which is due to the Hadamard product being a sub-matrix of the Kronecker product of the same matrices, the largest eigenvalue of the former are bounded by the largest eigenvalue of the latter, which is the product of the largest eigenvalues of the constituent matrices.

**Term 3:** Lastly, note that

$$
\begin{aligned}
2\left(\hat{\boldsymbol{\alpha}}\hat{\boldsymbol{\alpha}}^\top \odot (\hat{\mathbf{K}}^{-1} - \mathbf{I})\right) &= 2\mathbf{D}_{\hat{\boldsymbol{\alpha}}}(\hat{\mathbf{K}}^{-1} - \mathbf{I})\mathbf{D}_{\hat{\boldsymbol{\alpha}}} \\
&\succeq 2\mathbf{D}_{\hat{\boldsymbol{\alpha}}}(\lambda_{\min}(\hat{\mathbf{K}}^{-1})\mathbf{I} - \mathbf{I})\mathbf{D}_{\hat{\boldsymbol{\alpha}}} \\
&= 2(\lambda_{\min}(\hat{\mathbf{K}}^{-1}) - 1)\mathbf{D}_{\hat{\boldsymbol{\alpha}}}^2 \\
&= 2(\lambda_{\max}(\hat{\mathbf{K}})^{-1} - 1)\mathbf{D}_{\hat{\boldsymbol{\alpha}}}^2 \\
&\succeq 2(\lambda_{\min}(\mathbf{K})/\lambda_{\max}(\mathbf{K}) - 1)\mathbf{D}_{\hat{\boldsymbol{\alpha}}}^2 \\
&\succeq 2(\lambda_{\min}(\mathbf{K})/\lambda_{\max}(\mathbf{K}) - 1)\|\hat{\boldsymbol{\alpha}}\|_\infty^2\mathbf{I},
\end{aligned}
$$

where the last inequality comes from the second to last lower bound of Term 3 being non-positive, and therefore, being able to lower bound it with the largest magnitude entry of $\hat{\boldsymbol{\alpha}}$.

**Term 1 + 2 + 3:** Putting together the inequalities for all terms, we get

$$\lambda_{\min}(\mathbf{H_s}) \geq 2\lambda_{\max}(\hat{\mathbf{K}})^{-1} - \lambda_{\min}(\hat{\mathbf{K}})^{-2} + 2(\lambda_{\min}(\mathbf{K})/\lambda_{\max}(\mathbf{K}) - 1)\|\hat{\boldsymbol{\alpha}}\|_\infty^2,$$

where in slight abuse of notation, we let $\mathbf{H_s} = \mathbf{H_s}[-2\mathcal{L}]$ be the Hessian of the negative log likelihood.

Using the bound $\|\hat{\boldsymbol{\alpha}}\|_\infty \leq \|\hat{\boldsymbol{\alpha}}\|_2 \leq \lambda_{\min}(\hat{\mathbf{K}})^{-1}\|\hat{\mathbf{y}}\|$, where $\hat{\mathbf{y}} = \mathrm{diag}(\mathbf{K})^{-1/2}\mathbf{y}$. Therefore, $\|\hat{\boldsymbol{\alpha}}\|_\infty \leq \lambda_{\min}(\hat{\mathbf{K}})^{-1}\lambda_{\min}(\mathbf{K})^{-1/2}\|\mathbf{y}\|_2 \leq \lambda_{\max}(\mathbf{K})\lambda_{\min}(\mathbf{K})^{-3/2}\|\mathbf{y}\|_2$.

Finally, lower bounding the current lower bound by $m > 0$ yields a sufficient condition for the convexity at $\mathbf{s}$.

$$2\lambda_{\max}(\hat{\mathbf{K}})^{-1} - \lambda_{\min}(\hat{\mathbf{K}})^{-2} + 2(\lambda_{\min}(\mathbf{K})/\lambda_{\max}(\mathbf{K}) - 1)\lambda_{\min}(\hat{\mathbf{K}})^{-2}\lambda_{\min}(\mathbf{K})^{-1}\|\mathbf{y}\|_2^2 > m.$$

Re-arranging, we attain

$$\lambda_{\min}\hat{\lambda}_{\min}^2 \frac{(2\hat{\lambda}_{\max}^{-1} - \hat{\lambda}_{\min}^{-2} - m)}{2(1 - \lambda_{\min}/\lambda_{\max})} > \|\mathbf{y}\|_2^2, \tag{15}$$

which is a non-trivial guarantee when $2\hat{\lambda}_{\max}^{-1} - \hat{\lambda}_{\min}^{-2} - m > 0$. $\qquad\square$

**Lemma 13.** *[Smoothness via Eigenvalue Condition] Let $\hat{\mathbf{K}}$ as in Lemma 4. Suppose that $\|\mathbf{s}\|_\infty \leq s_{\max}$ and*

$$\lambda_{\min}\hat{\lambda}_{\min}^2 \frac{(M(1 - s_{\max})^2 + \hat{\lambda}_{\min}^{-2} - 2\hat{\lambda}_{\max}^{-1})}{2(\lambda_{\max}/\lambda_{\min} - 1)} > \|\mathbf{y}\|_2^2, \tag{16}$$

*where $\lambda_{\min,\max}$ (resp. $\hat{\lambda}_{\min,\max}$) are the smallest and largest eigenvalues of $\mathbf{K}$ (resp. $\hat{\mathbf{K}}$) respectively. Then $\mathbf{H_s} \prec M$. This is a non-trivial guarantee when $M(1 - s_{\max})^2 + \hat{\lambda}_{\min}^{-2} - 2\hat{\lambda}_{\max}^{-1} > 0$.*

*Proof.* We now derive an equivalent upper bound for the largest eigenvalue of the Hessian. Starting with the result of Lemma 4,

$$
\begin{aligned}
\mathbf{H_s}[-2\mathcal{L}(\boldsymbol{\rho}(\mathbf{s}))] &= \mathbf{D}_{1-\mathbf{s}}^{-1}\left[2\left(\hat{\boldsymbol{\alpha}}\hat{\boldsymbol{\alpha}}^\top \odot (\hat{\mathbf{K}}^{-1} - \mathbf{I})\right) + 2\,\mathrm{diag}(\hat{\mathbf{K}}^{-1}) - (\hat{\mathbf{K}}^{-1} \odot \hat{\mathbf{K}}^{-1})\right]\mathbf{D}_{1-\mathbf{s}}^{-1} \\
&\preceq \|(1-\mathbf{s})^{-1}\|_\infty^2 \left[2\left(\hat{\boldsymbol{\alpha}}\hat{\boldsymbol{\alpha}}^\top \odot (\hat{\mathbf{K}}^{-1} - \mathbf{I})\right) + 2\,\mathrm{diag}(\hat{\mathbf{K}}^{-1}) - (\hat{\mathbf{K}}^{-1} \odot \hat{\mathbf{K}}^{-1})\right].
\end{aligned}
$$

Therefore, it becomes immediately apparent that we will need to introduce an upper bound on $\|(1-\mathbf{s})^{-1}\|_\infty^2$, which is a restriction on the domain that $\boldsymbol{\rho}$ can take. Proceeding in a similar way as above, we bound the three terms in square brackets, now from above.

**Term 1:**

$$2\,\mathrm{diag}(\hat{\mathbf{K}}^{-1}) \preceq 2\lambda_{\max}(\hat{\mathbf{K}}^{-1})\mathbf{I} = 2\lambda_{\min}(\hat{\mathbf{K}})^{-1}\mathbf{I}.$$

**Term 2:**

$$-(\hat{\mathbf{K}}^{-1} \odot \hat{\mathbf{K}}^{-1}) \preceq -\lambda_{\min}(\hat{\mathbf{K}}^{-1})^2\mathbf{I} = -\lambda_{\max}(\hat{\mathbf{K}})^{-2}\mathbf{I}.$$

**Term 3:**

$$2\left(\hat{\boldsymbol{\alpha}}\hat{\boldsymbol{\alpha}}^\top \odot (\hat{\mathbf{K}}^{-1} - \mathbf{I})\right) = 2\mathbf{D}_{\hat{\boldsymbol{\alpha}}}(\hat{\mathbf{K}}^{-1} - \mathbf{I})\mathbf{D}_{\hat{\boldsymbol{\alpha}}}$$

$$\preceq 2\mathbf{D}_{\hat{\boldsymbol{\alpha}}}(\lambda_{\max}(\hat{\mathbf{K}}^{-1})\mathbf{I} - \mathbf{I})\mathbf{D}_{\hat{\boldsymbol{\alpha}}}$$

$$= 2(\lambda_{\max}(\hat{\mathbf{K}}^{-1}) - 1)\mathbf{D}_{\hat{\boldsymbol{\alpha}}}^2$$

$$= 2(\lambda_{\min}(\hat{\mathbf{K}})^{-1} - 1)\mathbf{D}_{\hat{\boldsymbol{\alpha}}}^2$$

$$\preceq 2(\lambda_{\max}(\mathbf{K})/\lambda_{\min}(\mathbf{K}) - 1)\mathbf{D}_{\hat{\boldsymbol{\alpha}}}^2$$

$$\preceq 2(\lambda_{\max}(\mathbf{K})/\lambda_{\min}(\mathbf{K}) - 1)\|\hat{\boldsymbol{\alpha}}\|_\infty^2\mathbf{I},$$

where now the last inequality follows because the second to last expression is always non-negative.
**Term 1 + 2 + 3:** Putting together the inequalities for all terms, we get

$$\lambda_{\max}(\mathbf{H}_\mathbf{s}) \leq 2\lambda_{\min}(\hat{\mathbf{K}})^{-1} - \lambda_{\max}(\hat{\mathbf{K}})^{-2} + 2(\lambda_{\max}(\mathbf{K})/\lambda_{\min}(\mathbf{K}) - 1)\|\hat{\boldsymbol{\alpha}}\|_\infty^2,$$

Finally, upper bounding the current upper bound by $M/\|(1-\mathbf{s})^{-1}\|_\infty^2 > 0$ yields a sufficient condition for the $M$-smoothness at $\mathbf{s}$. Using the same bound for $\|\hat{\boldsymbol{\alpha}}\|_\infty$ derived for the convexity result,

$$2\lambda_{\min}(\hat{\mathbf{K}})^{-1} - \lambda_{\max}(\hat{\mathbf{K}})^{-2} + 2((\lambda_{\max}(\mathbf{K})/\lambda_{\min}(\mathbf{K}) - 1)\lambda_{\min}(\hat{\mathbf{K}})^{-2}\lambda_{\min}(\mathbf{K})^{-1}\|\mathbf{y}\|_2^2 < M.$$

Re-arranging, we attain

$$\lambda_{\min}\hat{\lambda}_{\min}^2 \frac{(M(1-s_{\max})^2 + \hat{\lambda}_{\min}^{-2} - 2\hat{\lambda}_{\max}^{-1})}{2(\lambda_{\max}/\lambda_{\min} - 1)} > \|\mathbf{y}\|_2^2, \tag{17}$$

which is a non-trivial guarantee when $M(1-s_{\max})^2 + \hat{\lambda}_{\min}^{-2} - 2\hat{\lambda}_{\max}^{-1} > 0$. $\qquad\square$

**Lemma 7.** *[Strong Convexity via Diagonal Dominance] Let $m > 0$ and $\mathbf{K}_0$ be $\delta$-diagonally dominant with $\delta < \left((5-m) - \sqrt{25 - 9m + 17}\right)/4 \leq (5 - \sqrt{17})/4 \approx 0.44$ and*

$$\lambda_{\min}(\mathbf{K}_0)(1-\delta)^2\frac{2(1+\delta)^{-1} - (1-\delta)^{-2} - m}{2(1 - (1-\delta)/(1+\delta))} \geq \|\mathbf{y}\|_2^2.$$

*Then the NMLL is $m$-strongly convex for all $\mathbf{s} \in [0,1]^n$, i.e. $\boldsymbol{\rho}(\mathbf{s}) \in [0,\infty]^n$.*

*Proof.* Fist, Gershgorin's Disk Theorem implies that the eigenvalues of $\mathrm{diag}(\mathbf{A})^{-1/2}\mathbf{A}\,\mathrm{diag}(\mathbf{A})^{-1/2}$ lie in $(1-\delta, 1+\delta)$ for a $\delta$-diagonally dominant matrix $\mathbf{A}$. Further, the condition number of $\mathbf{A}$ is bounded above by $\kappa(\mathbf{A}) = \lambda_{\max}(\mathbf{A})/\lambda_{\min}(\mathbf{A}) \leq (1+\delta)/(1-\delta)$. See Horn and Johnson [30] for more background on matrix analysis. Plugging these bounds into the results of Lemma 5 yields

$$\lambda_{\min}(\mathbf{K})\hat{\lambda}_{\min}^2\frac{(2\hat{\lambda}_{\max}^{-1} - \hat{\lambda}_{\min}^{-2} - m)}{2(1 - \lambda_{\min}/\lambda_{\max})} \geq \lambda_{\min}(\mathbf{K})(1-\delta)^2\frac{(2(1+\delta)^{-1} - (1-\delta)^{-2} - m)}{2(1 - (1-\delta)/(1+\delta))}.$$

Lower bounding the last expression by $\|\mathbf{y}\|_2^2$ implies $m$-strong convexity. This gives rise to a non-trivial guarantee whenever the numerator is larger than zero. In particular,

$$2(1+\delta)^{-1} - (1-\delta)^{-2} - m > 0.$$

Expanding, noting that $0 < \delta < 1$ implies $\delta^3 < \delta^2$ in order to reduce a power in the resulting expression, and collecting terms with like powers, we attain the following sufficient condition

$$2\delta^2 + (m-5)\delta + (1-m) > 0.$$

Note that for this condition to hold if $\delta = 0$, we need to have $m < 1$. Fortunately, this is a quadratic in $\delta$ whose smallest positive root is

$$\delta_- = \frac{1}{4}\left((5-m) - \sqrt{(5-m)^2 - 8(1-m)}\right).$$

In particular, for $m = 0$, this reduces to

$$\delta_- = \frac{1}{4}\left(5 - \sqrt{17}\right) \approx 0.4384471871911697.$$

Lastly, note that if $\mathbf{K}_0$ is $\delta$-diagonally dominant, then so is $\mathbf{K_s} = \mathbf{K}_0 + \mathbf{D}_{\boldsymbol{\rho}}(\mathbf{s})$, since the robust variances add to the diagonal, making it more dominant. Therefore, the convexity guarantee holds for all $\boldsymbol{\rho}(\mathbf{s})$, if it holds for the base covariance matrix $\mathbf{K}_0$. Note that $\lambda_{\min}(\mathbf{K}) \geq \sigma^2$. $\qquad \square$

Similarly, we can prove a similar statement relating diagonal dominance with $M$-smoothness.

**Lemma 14.** *[Smoothness via Diagonal Dominance] Suppose $\mathbf{K}_0$ is a $\delta$-diagonally dominant covariance matrix and suppose we constrain $\|\mathbf{s}\|_\infty \leq s_{\max} \leq 1 - \sqrt{1/M}$. Then*

$$\lambda_{\min}(\mathbf{K}_0)(1 - \delta)^2 \frac{M(1 - s_{\max})^2 - 1}{2((1 + \delta)/(1 - \delta) - 1)} \geq \|\mathbf{y}\|_2^2,$$

*implies that the NLML is $m$-strongly convex for all $\mathbf{s} \in [0, 1]^n$, i.e. $\boldsymbol{\rho}(\mathbf{s}) \in [0, \infty]^n$.*

*Proof.* We proceed in a similar way as for Lemma 7, but with Lemma 13 as the starting point.

$$\lambda_{\min}(\mathbf{K})(\hat{\lambda}_{\min})^2 \frac{(M(1 - s_{\max})^2 + \hat{\lambda}_{\min}^{-2} - 2\hat{\lambda}_{\max}^{-1})}{2(\lambda_{\max}/\lambda_{\min} - 1)} \geq \lambda_{\min}(\mathbf{K})(1 - \delta)^2 \frac{M(1 - s_{\max})^2 - 1}{2((1 + \delta)/(1 - \delta) - 1)},$$

where we used $(\hat{\lambda}_{\min}^{-2} - 2\hat{\lambda}_{\max}^{-1}) \geq -1$, which is tight when $\hat{\mathbf{K}}$ is unitary. Lower bounding the last expression by $\|\mathbf{y}\|_2^2$ implies $M$-smoothness. This gives rise to a non-trivial guarantee whenever the numerator is larger than zero. In particular, $M(1 - s_{\max})^2 - 1 > 0$, which implies $s_{\max} \leq 1 - \sqrt{1/M}$ or equivalently, $M > 1/(1 - s_{\max})^2$. Lastly, note that if $\mathbf{K}_0$ is $\delta$-diagonally dominant, then so is $\mathbf{K_s} = \mathbf{K}_0 + \mathbf{D}_{\boldsymbol{\rho}}(\mathbf{s})$, since the robust variances only add to the diagonal. Therefore, if the inequality holds for the base covariance matrix $\mathbf{K}_0$, the smoothness guarantee holds for all $\boldsymbol{\rho}(\mathbf{s})$ such that $\mathbf{s} \leq s_{\max} \leq 1 - \sqrt{1/M}$. Note also that $\lambda_{\min}(\mathbf{K}) \geq \sigma^2$. $\qquad \square$

**Theorem 8.** *[Approximation Guarantee] Let $\mathbf{K}_0 = k(\mathbf{X}, \mathbf{X}) + \sigma^2 \mathbf{I}$ be $\delta$-diagonally dominant, $s_{\max} > 0$ be an upper bound on $\|\mathbf{s}\|_\infty$, and suppose $\|\mathbf{y}\|, \delta$ satisfy the bounds of Lemmas 7 and 14, guaranteeing $m$-convexity and $M$-smoothness of the NMLL for some $m > 0$, $M > 1/(1 - s_{\max})^2$. Let $\mathbf{s}_{\mathrm{OMP}}(r)$ be the $r$-sparse vector attained by OMP on the NMLL objective for $r$ steps, and let $\mathbf{s}_{\mathrm{OPT}}(r) = \arg\max_{\|\mathbf{s}\|_0 = r, \|\mathbf{s}\|_\infty \leq s_{\max}} \mathcal{L}(\boldsymbol{\rho}(\mathbf{s}))$ be the optimal $r$-sparse vector. Then for any $2r \leq n$,*

$$\tilde{\mathcal{L}}\left(\boldsymbol{\rho}(\mathbf{s}_{\mathrm{OMP}}(r))\right) \geq \left(1 - e^{-m/M}\right) \tilde{\mathcal{L}}\left(\boldsymbol{\rho}(\mathbf{s}_{\mathrm{OPT}}(r))\right),$$

*where $\tilde{\mathcal{L}}(\cdot) = \mathcal{L}(\cdot) - \mathcal{L}(\mathbf{0})$ is normalized so that $\max_{\mathbf{s}_\mathcal{S}} \tilde{\mathcal{L}}(\mathbf{s}_\mathcal{S}) \geq 0$ for any support $\mathcal{S}$.*

*Proof.* The result is a direct consequence of meeting the $m$-convexity and $M$-smoothness conditions of Lemmas 7, 14 above, and the OMP approximation guarantee of Theorem 11 due to Elenberg et al. [23]. Note that the condition on $2r \leq n$ comes from the RSC condition in Theorem 11 being required for subsets of size $2r$. As we proved bounds for the $m$-convexity of the full Hessian of size $n$, $r$ has to be smaller than $n/2$ for the assumptions of the theorem to hold. Regarding the upper bound $s_{\max}$ on $\mathbf{s}$, we note that the constraint is convex and therefore doesn't change the convexity property of the optimization problem.

Further, note that $\max_{\boldsymbol{\rho}_\mathcal{S}} \mathcal{L}(\boldsymbol{\rho}_\mathcal{S}) \leq \max_{\boldsymbol{\rho}_{\mathcal{S} \cup i}} \mathcal{L}(\boldsymbol{\rho}_{\mathcal{S} \cup i})$, since the additional non-zero element could stay at 0, if the marginal likelihood does not improve with $\rho_i$ increasing. That is, the subset selection problem is monotonic. As a consequence, we can normalize the MLL by $\tilde{\mathcal{L}}(\cdot) = \mathcal{L}(\cdot) - \mathcal{L}(\mathbf{0})$, which then only takes positive values for any $\mathbf{s}_\mathcal{S}^* = \arg\max_{\mathbf{s}\setminus\mathcal{S}=0} \mathcal{L}(\boldsymbol{\rho}(\mathbf{s}_\mathcal{S}))$, i.e. $\max_{\mathbf{s}_\mathcal{S}} \tilde{\mathcal{L}}(\mathbf{s}_\mathcal{S}) \geq 0$. This normalization is required for the constant factor approximation guarantee to apply, similar to the original work of Nemhauser. $\qquad \square$

This theoretical approach could lead to approximation guarantees for Tipping [60]'s Sparse Bayesian Learning (SBL) model, for which Ament and Gomes [9] show that greedily optimizing the associated NMLL is equivalent to stepwise regression in the limit of $\sigma \to 0$, proving exact recovery guarantees.

# D  Additional Detail on the Experiments

Our benchmark uses a modified version of the code from Andrade and Takeda [10], available at `https://github.com/andrade-stats/TrimmedMarginalLikelihoodGP` under the GNU GPLv2 license.

## D.1  Synthetic Regression Problems

**Model fitting runtimes**  Fig. 8 summarizes the fitting times for the different models on the different scenarios from Section 6.1. We observe that the outlier type and the fraction of outliers both have relatively limited effect on the fitting times for all of the models. Fig. 9 therefore provides a more compact view of the same data (aggregated across outlier types and outlier fractions), giving a better sense of the distribution of the fitting times. Unsurprisingly, the baselines that simply fit a single GP model ("vanilla", "winsorize", "power_transform") are substantially faster than any of the robust approaches. While all of the robust models show similar fitting times on the Hartmann problem, fitting our RRP is significantly faster (note the logarithmic scale of the y-axis) than fitting the Student-$t$ and trimmed MLE models on the 5-dim and 10-dim Friedman problems. The trimmed MLE model in particular ends up being quite slow, especially on the 5d Friedman function.

## D.2  BO experiments, additional details

For Hartmann6, we consider the standard domain of $[0, 1]^6$.

**SVM**  The SVM problem, the goal is to optimize the test RMSE of an SVM regression model trained on 100 features from the CT slice UCI dataset. We tune the following three parameters: $C \in$ [1e-2, 1e2], $\epsilon \in$ [1e-2, 1], and $\gamma \in$ [1e-3, 1e-1]. All parameters are tuned in log-scale. Corruptions simulate I/O failures in which case we only train on $U[100, 1000]$ training points out of the available $50,000$ training observations.

**CNN**  For the 5D CNN problem the goal is to optimize the test accuracy of a CNN classifier trained on the MNIST dataset. We tune the following 5 parameters: learning rate in the interval [1e-4, 1e-1], momentum in the interval [0, 1], weight decay in the interval [0, 1], step size in the interval [1, 100], and $\gamma \in$ [0, 1]. Similarly to the SVM problem, we only train on $U[100, 1000]$ of the available training batches when an I/O failure occurs.

**Rover**  The rover trajectory planning problem was originally proposed in Wang et al. [65]. The goal is to tune the trajectory of a rover in order to maximize the reward of its final trajectory. We use the same obstacle locations and trajectory parameterization as in Maus et al. [47], with the main difference being that we parameterize the trajectory using 10 points, resulting in 20 tunable parameters. When a corruption occurs, the rover will stop at a uniformly random point along its trajectory, generally resulting in lower reward than the original trajectory.

## D.3  Additional Synthetic BO Results

In addition to the results in Fig. 6, we also include results when the corruption probability is 20% in Fig. 10. We observe that the results are similar as with 10% corruption probability, but that the performance of several baselines regresses significantly.

## D.4  Computational Setup and Requirements

Robust GP regression is very data-efficient, focuses on the small-data regime, and runs fast (faster than competing baselines studied in this paper). Therefore, each individual run required very limited compute resources (this includes the baseline methods). To produce statistically meaningful results, however, we ran a large number of replications for both our regression and Bayesian optimization benchmarks on a proprietary cluster. We estimate the amount of compute spent on these experiments to be around 2 CPU years in total, using standard (Intel Xeon) CPU hardware. The amount of compute spent on exploratory investigations as part of this work was negligible (this was ad-hoc exploratory and development work on a single CPU machine).

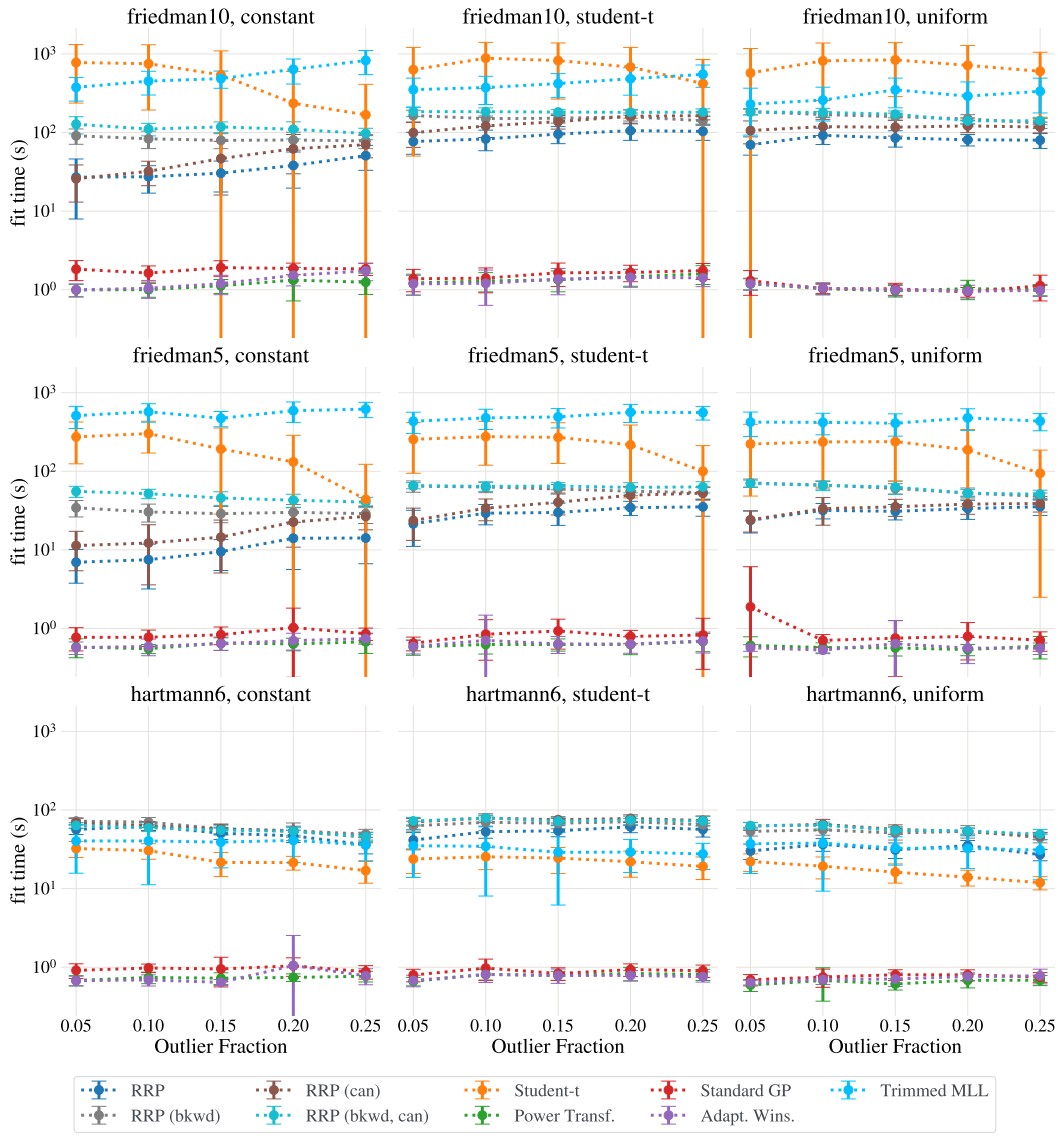

Figure 8: Fitting times of the different robust GP modeling approaches. The plots show means and one standard deviation. Here "bkwd" indicates the backward variant of RRP (Algorithm 2), and "can" indicates the canonical (non-convex) parameterization.

## D.5 Impact of Convex Parameterization on Joint Optimization of Hyper-Parameters

A limitation of Lemma 4 is that it only guarantees convexity for the sub-problem of optimizing the $\rho_i$'s. In practice, we jointly optimize all GP hyper-parameters, including length scales. In this case, the theory guarantees the positive-definiteness of the *submatrix* of the Hessian corresponding to $\boldsymbol{\rho}(\mathbf{s})$, and we expect this to improve the quality of the results of the numerical optimization routines.

Indeed, positive-definiteness is beneficial for quasi-Newton optimization algorithms like L-BFGS [42], which restarts its approximation to the Hessian whenever it encounters non-convex regions, because the associated updates to the Hessian approximation are not positive-definite. This leads the algorithm to momentarily revert back to gradient descent, with an associated slower convergence rate.

To quantify the impact of this, we ran convergence analyses using the data from Fig. 1, allowing all $\boldsymbol{\rho}$ to be optimized jointly with other hyper-parameters (length scale, kernel variance and noise

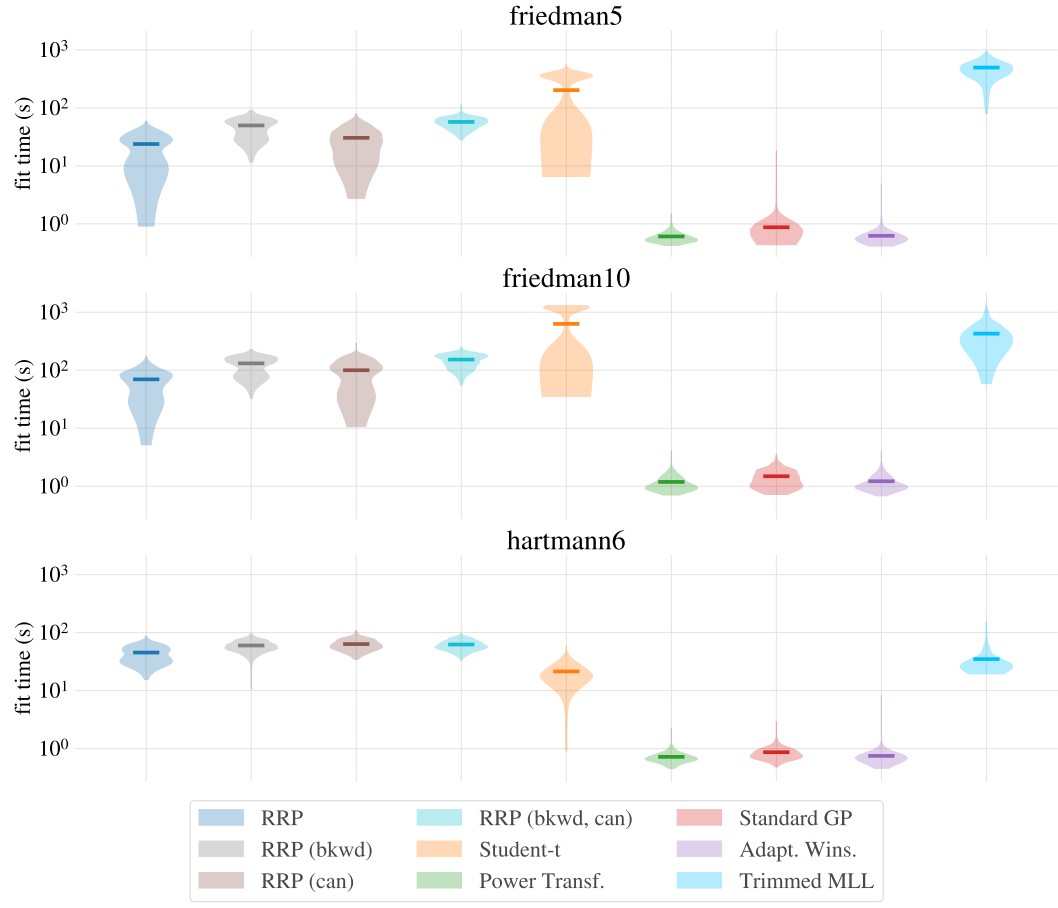

Figure 9: Fitting times of the different robust GP modeling approaches on the regression tasks from Section 6.1. Results are aggregated across outlier types and outlier fractions as those do not affect fitting times much (see Fig. 8).

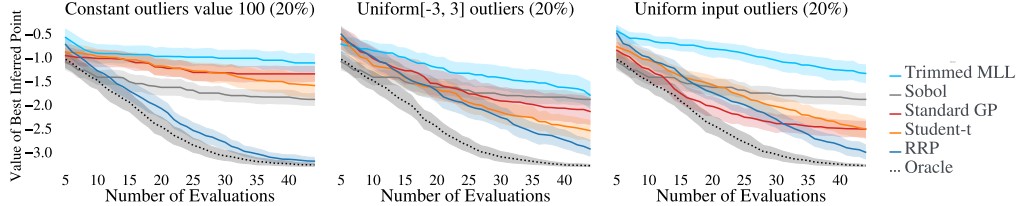

Figure 10: BO results for Hartmann6: *Left:* We see that Relevance pursuit performs well in the case of constant outliers of value 100 and almost performs just as well as the oracle. *Middle:* Relevance pursuit performs the best followed by the Student-$t$ likelihood from [46] in the case of $U[-3, 3]$. No method performs as well as the oracle when the outlier probability is 20%. *Right:* Similarly to the middle column, this setting hides the outliers within the range of the outliers making it difficult to match the performance of the oracle.

variance), recording the achieved negative log marginal likelihood (NLML) as a function of the tolerance parameter ftol of the L-BFGS optimizer. The results are reported in Table D.5, and indicate that the optimizer terminates with a much better NLML using the convex parameterization with the same convergence tolerance.

There are also settings in which we do not actually jointly optimize the hyper-parameters, particularly when we have access to data from the same data generating process that has been manually labeled

| ftol | Canonical $\rho$ | Convex $\rho(\mathrm{s})$ |
|------|------------------|---------------------------|
| 1e−3 | −4.37 | **−14.18** |
| 1e−4 | −4.37 | **−93.00** |
| 1e−5 | −4.37 | **−93.01** |
| 1e−6 | −4.37 | **−135.05** |
| 1e−7 | −98.62 | **−518.68** |
| 1e−8 | −97.52 | **−1139.29** |

Table 1: Comparison of the negative log marginal likelihood achieved after numerical optimization of the canonical and convex parameterization of $\rho$ with L-BFGS. Notably, the convex parameterization leads to improved marginal likelihood values for for a given convergence tolerance `ftol`.

by domain experts as outlier-free. Then we can estimate the model hyper-parameters on that data, and fix them for the RRP on new data sets that we do not know to be outlier-free.

### D.6 Comparison to Robust and Conjugate Gaussian Processes (RCGP)

We report extended empirical comparisons with Altamirano et al. [2]'s RCGP method, using their experimental setup and method implementation in GPFLow. Including GPFlow in our own benchmarking setup and compute resources proved difficult. To circumvent this, we wrote wrappers for both BoTorch's standard GP and RRP, which also accounts for any orthogonal implementation differences between the two frameworks, and ran the benchmarks locally on an M-series MacBook.

See Tables 2 and 3 for the mean absolute error and negative log predictive density, respectively. The tables include the empirical mean and standard deviation over 20 replications on corrupted version of the following base data sets: 1) Synthetic, which is generated as a draw of a GP with a exponentiated quadratic kernel, and four data sets available on the UCI machine learning repository[34], in particular, 2) Boston [29], 3) Concrete [68], 4) Energy [63], and 5) Yacht [27]. The benchmark considers no corruptions ("No Outliers"), "Asymmetric Outliers", which are uniform in $x$ are shifted negatively in $y$, "Uniform Outliers", which shift $y$ in both directions (positively and negatively), and "Focused Outliers", which form concentrated clusters in both $x$ and $y$. Any bold entry in the table signifies a results that is within one standard-error of the best result's one standard-error confidence bound.

Table 2: Mean absolute error (MAE) using Altamirano et al. [2]'s experimental setup in GPFlow. RRP is always competitive with the other methods, and outperforms them significantly for uniform and asymmetric outliers.

| | Standard GP (GPFlow) | Student-$t$ GP (GPFlow) | RCGP (GPFlow) | Standard GP (BoTorch) | RRP (BoTorch) |
|---|---|---|---|---|---|
| **No Outliers** | | | | | |
| Synthetic | **8.82e-2 (2.12e-3)** | **8.81e-2 (2.09e-3)** | **8.77e-2 (2.08e-3)** | **8.90e-2 (2.19e-3)** | **8.90e-2 (2.19e-3)** |
| Boston | 2.24e-1 (7.11e-3) | 2.24e-1 (6.50e-3) | 2.24e-1 (7.29e-3) | **2.08e-1 (5.53e-3)** | **2.08e-1 (5.75e-3)** |
| Concrete | 2.00e-1 (3.72e-3) | **1.99e-1 (3.73e-3)** | **1.93e-1 (4.05e-3)** | **1.92e-1 (3.80e-3)** | **1.93e-1 (3.66e-3)** |
| Energy | 3.01e-2 (8.65e-4) | 3.44e-2 (1.46e-3) | **2.42e-2 (1.01e-3)** | 3.09e-2 (9.41e-4) | 3.03e-2 (8.88e-4) |
| Yacht | **1.69e-2 (2.09e-3)** | 2.02e-2 (1.29e-3) | 2.19e-2 (4.63e-3) | **1.49e-2 (1.95e-3)** | **1.42e-2 (1.58e-3)** |
| **Uniform Outliers** | | | | | |
| Synthetic | 3.45e-1 (1.48e-2) | 2.93e-1 (1.23e-2) | 2.15e-1 (9.30e-3) | 3.48e-1 (1.62e-2) | **8.99e-2 (2.37e-3)** |
| Boston | 4.97e-1 (2.80e-2) | 3.86e-1 (1.52e-2) | 4.94e-1 (1.81e-2) | 6.81e-1 (2.25e-2) | **2.83e-1 (3.16e-2)** |
| Concrete | 3.80e-1 (8.36e-3) | 3.54e-1 (5.98e-3) | 4.43e-1 (7.97e-3) | 3.80e-1 (7.77e-3) | **2.03e-1 (3.98e-3)** |
| Energy | 2.67e-1 (8.93e-3) | 2.73e-1 (2.40e-2) | 3.05e-1 (3.55e-2) | 5.60e-1 (1.47e-2) | **3.06e-2 (8.35e-4)** |
| Yacht | 3.23e-1 (1.68e-2) | 2.29e-1 (1.54e-2) | 4.18e-1 (3.48e-2) | 6.81e-1 (3.21e-2) | **1.20e-2 (1.04e-3)** |
| **Asymmetric Outliers** | | | | | |
| Synthetic | 1.11e+0 (2.24e-2) | 1.02e+0 (2.34e-2) | 7.72e-1 (3.31e-2) | 1.17e+0 (2.55e-2) | **2.71e-1 (9.96e-2)** |
| Boston | 6.92e-1 (1.40e-2) | 5.80e-1 (1.24e-2) | 5.31e-1 (2.33e-2) | 8.55e-1 (3.04e-2) | **3.35e-1 (4.33e-2)** |
| Concrete | 6.60e-1 (1.21e-2) | 5.55e-1 (1.08e-2) | 4.91e-1 (1.11e-2) | 6.60e-1 (1.25e-2) | **2.03e-1 (4.45e-3)** |
| Energy | 6.03e-1 (1.12e-2) | 4.74e-1 (9.99e-3) | 4.08e-1 (3.34e-2) | 7.05e-1 (1.78e-2) | **3.07e-2 (8.33e-4)** |
| Yacht | 5.90e-1 (1.60e-2) | 4.58e-1 (9.31e-3) | 4.32e-1 (3.30e-2) | 8.36e-1 (3.76e-2) | **2.70e-2 (1.35e-2)** |
| **Focused Outliers** | | | | | |
| Synthetic | **1.89e-1 (1.50e-2)** | **1.92e-1 (1.56e-2)** | **1.64e-1 (1.29e-2)** | **1.80e-1 (1.46e-2)** | 2.00e-1 (1.62e-2) |
| Boston | **2.44e-1 (8.48e-3)** | **2.60e-1 (1.30e-2)** | **2.49e-1 (1.04e-2)** | **2.49e-1 (8.37e-3)** | **2.48e-1 (8.38e-3)** |
| Concrete | 2.40e-1 (5.54e-3) | **2.35e-1 (5.63e-3)** | **2.38e-1 (1.10e-2)** | **2.24e-1 (6.20e-3)** | **2.25e-1 (5.86e-3)** |
| Energy | 8.83e-2 (5.46e-2) | 3.24e-2 (1.36e-3) | **2.92e-2 (8.95e-4)** | **3.03e-2 (8.95e-4)** | **3.03e-2 (8.95e-4)** |
| Yacht | 2.42e-1 (6.89e-2) | 1.10e-1 (2.07e-2) | **1.93e-2 (1.88e-3)** | **1.78e-2 (2.96e-3)** | **1.59e-2 (2.20e-3)** |

### D.7 Comparison to Robust Gaussian Process with Huber Likelihood

In the following, we compare our method with additional variational GP baselines with Laplace and Huber likelihoods, and translated the Matlab code of the "projection statistics" of Algikar and Mili [1] to PyTorch. We then combined the projection-statistics-based weighting of the Huber loss with

Table 3: Negative log predictive density (NLPD) using Altamirano et al. [2]'s experimental setup in GPFlow. RRP is generally competitive, and outperforms other methods significantly for uniform and asymmetric outliers.

|  | Standard GP (GPFlow) | Student-$t$ GP (GPFLow) | RCGP (GPFlow) | Standard GP (BoTorch) | RRP (BoTorch) |
|---|---|---|---|---|---|
| No Outliers |  |  |  |  |  |
| Synthetic | **-8.21e-1 (2.16e-2)** | **-8.20e-1 (2.16e-2)** | **-8.23e-1 (2.21e-2)** | **-8.05e-1 (2.25e-2)** | **-8.05e-1 (2.24e-2)** |
| Boston | 2.08e-1 (4.26e-2) | 1.99e-1 (3.76e-2) | 1.95e-1 (4.69e-2) | **9.24e-2 (2.82e-2)** | **9.43e-2 (2.82e-2)** |
| Concrete | **1.48e-1 (2.33e-2)** | **1.29e-1 (2.31e-2)** | **1.09e-1 (3.19e-2)** | **1.11e-1 (2.73e-2)** | **1.16e-1 (2.68e-2)** |
| Energy | -1.72e+0 (3.83e-2) | -1.62e+0 (3.46e-2) | **-1.96e+0 (4.37e-2)** | -1.67e+0 (3.08e-2) | -1.69e+0 (4.02e-2) |
| Yacht | -1.79e+0 (3.23e-1) | -2.05e+0 (6.61e-2) | -2.00e+0 (3.27e-1) | **-2.47e+0 (1.06e-1)** | -2.23e+0 (1.84e-1) |
| Uniform Outliers |  |  |  |  |  |
| Synthetic | 1.60e+0 (1.41e-2) | 1.47e+0 (1.55e-2) | 1.57e+0 (1.39e-2) | 1.58e+0 (1.40e-2) | **-7.99e-1 (2.26e-2)** |
| Boston | 1.67e+0 (7.55e-3) | 1.54e+0 (8.98e-3) | 1.64e+0 (1.12e-2) | 1.81e+0 (5.95e-2) | **4.52e-1 (1.34e-1)** |
| Concrete | 1.64e+0 (6.83e-3) | 1.51e+0 (6.56e-3) | 1.63e+0 (7.57e-3) | 1.63e+0 (6.72e-3) | **1.47e-1 (2.51e-2)** |
| Energy | 1.61e+0 (7.75e-3) | 1.50e+0 (1.76e-2) | 1.61e+0 (9.74e-3) | 1.70e+0 (8.58e-3) | **-1.67e+0 (3.72e-2)** |
| Yacht | 1.61e+0 (9.50e-3) | 1.46e+0 (1.19e-2) | 1.62e+0 (1.87e-2) | 1.78e+0 (4.67e-2) | **-2.32e+0 (1.97e-1)** |
| Asymmetric Outliers |  |  |  |  |  |
| Synthetic | 1.94e+0 (9.39e-3) | 1.90e+0 (9.98e-3) | 1.89e+0 (1.13e-2) | 1.93e+0 (1.06e-2) | **-3.70e-1 (2.30e-1)** |
| Boston | 1.68e+0 (1.01e-2) | 1.56e+0 (1.09e-2) | 1.64e+0 (9.51e-3) | 2.52e+0 (7.26e-1) | **6.14e-1 (1.19e-1)** |
| Concrete | 1.66e+0 (7.43e-3) | 1.54e+0 (7.45e-3) | 1.62e+0 (6.63e-3) | 1.65e+0 (7.55e-3) | **1.55e-1 (2.86e-2)** |
| Energy | 1.62e+0 (8.85e-3) | 1.49e+0 (9.70e-3) | 1.62e+0 (1.82e-2) | 1.71e+0 (9.79e-3) | **-1.66e+0 (3.55e-2)** |
| Yacht | **1.59e+0 (9.35e-3)** | **1.45e+0 (1.01e-2)** | **1.56e+0 (8.15e-3)** | 1.75e+0 (1.41e-2) | **-2.17e-1 (1.82e+0)** |
| Focused Outliers |  |  |  |  |  |
| Synthetic | 6.78e-1 (6.59e-2) | **6.11e-1 (8.68e-2)** | **5.69e-1 (4.07e-2)** | **6.66e-1 (6.60e-2)** | 9.67e+0 (2.24e+0) |
| Boston | **2.57e-1 (4.23e-2)** | 3.21e-1 (6.40e-2) | **2.74e-1 (6.13e-2)** | **1.98e-1 (3.04e-2)** | **1.97e-1 (3.06e-2)** |
| Concrete | 2.77e-1 (2.58e-2) | **2.72e-1 (2.81e-2)** | 2.43e-1 (4.26e-2) | 2.22e-1 (2.87e-2) | 2.28e-1 (2.72e-2) |
| Energy | 2.81e+4 (2.81e+4) | **-1.67e+0 (4.13e-2)** | **-1.74e+0 (4.98e-2)** | **-1.70e+0 (4.59e-2)** | **-1.70e+0 (4.59e-2)** |
| Yacht | 9.38e+4 (5.09e+4) | -4.60e-1 (2.91e-1) | -2.47e+0 (8.09e-2) | **-2.63e+0 (5.55e-2)** | **-2.61e+0 (5.66e-2)** |

a variational (referred to as Huber-Projection) to get as close as possible to a direct comparison to Algikar and Mili [1] without access to a Matlab license.

Tables 4 and 5 shows the root mean square error and negative log predictive density on the Neal, Friedman 5 and Friedman 10 test functions, as well as the Yacht Hydrodynamics [27] and California Housing [51] datasets from the UCI database [34], where 15% of the training data sets of the models were corrupted. Tables 6 and 7 below were generated in a similar way, but 100% of the data were subject to heavier-tailed noise, either Student-$t$ or Laplace.

In summary, the Relevance Pursuit model generally outperforms the variational GPs with heavy-tailed likelihoods when the corruptions are a sparse subset of all observations. Unsurprisingly, the GPs with heavy-tailed likelihoods perform best when *all* observations are subject to heavy-tailed noise. While such uniformly heavy-tailed noise does exist in practice, we stress that this is a distinct setting to the common setting where datasets contain a *subset* of a-priori unknown outliers, while a dominant fraction of the data can be considered inliers that, once they are identified, can be used to train a model without additional treatment.

Table 4: Comparison with Huber GP: Root mean square error with 15% Corruptions

| Data | Standard | Relevance Pursuit | Student-$t$ | Laplace | Huber | Huber + Projection |
|---|---|---|---|---|---|---|
| **Neal** | | | | | | |
| Uniform | 3.87e-1 (2.40e-2) | **3.79e-2 (1.10e-2)** | 4.18e-1 (8.65e-3) | 4.18e-1 (4.72e-3) | 4.18e-1 (4.76e-3) | 4.18e-1 (4.76e-3) |
| Constant | 1.91e+0 (1.18e-1) | **4.37e-2 (1.87e-2)** | 7.26e-1 (1.38e-1) | 4.73e-1 (1.28e-2) | 4.89e-1 (1.62e-2) | 4.89e-1 (1.62e-2) |
| Student-$t$ | 7.74e-1 (2.70e-1) | **6.71e-2 (2.29e-2)** | 3.90e-1 (3.59e-3) | 4.07e-1 (4.79e-3) | 4.09e-1 (5.02e-3) | 4.09e-1 (5.02e-3) |
| Laplace | 7.19e-1 (6.17e-2) | **6.27e-2 (1.52e-2)** | 4.01e-1 (4.42e-3) | 4.20e-1 (4.34e-3) | 4.20e-1 (4.00e-3) | 4.20e-1 (4.00e-3) |
| **Friedman 5** | | | | | | |
| Uniform | 2.05e+0 (8.83e-2) | **9.40e-2 (1.23e-2)** | 7.01e-1 (1.37e-1) | 7.30e-1 (7.33e-2) | 6.78e-1 (6.67e-2) | 7.12e-1 (7.50e-2) |
| Constant | 1.48e+1 (5.16e-1) | **7.29e-2 (5.40e-3)** | 8.98e-1 (1.40e-1) | 2.37e+0 (5.19e-2) | 2.50e+0 (7.18e-2) | 2.50e+0 (7.18e-2) |
| Student-$t$ | 7.91e+0 (8.80e-1) | **1.17e-1 (2.24e-2)** | 5.26e-1 (1.26e-1) | 1.27e+0 (9.49e-2) | 1.36e+0 (9.34e-2) | 1.31e+0 (9.73e-2) |
| Laplace | 7.59e+0 (4.98e-1) | **9.63e-2 (1.15e-2)** | 6.62e-1 (1.34e-1) | 1.51e+0 (7.78e-2) | 1.63e+0 (6.89e-2) | 1.63e+0 (6.89e-2) |
| **Friedman 10** | | | | | | |
| Uniform | 1.67e+0 (4.94e-2) | **4.84e-2 (6.21e-3)** | 5.92e-1 (1.36e-1) | 6.82e-1 (1.03e-1) | 6.74e-1 (1.04e-1) | 5.93e-1 (9.43e-2) |
| Constant | 1.34e+1 (2.97e-1) | **5.12e-2 (2.69e-3)** | 8.63e-1 (1.39e-1) | 2.10e+0 (2.39e-2) | 2.18e+0 (3.01e-2) | 2.18e+0 (3.01e-2) |
| Student-$t$ | 7.93e+0 (1.75e+0) | **8.45e-2 (2.42e-2)** | 5.30e-1 (1.30e-1) | 1.40e+0 (9.10e-2) | 1.40e+0 (9.12e-2) | 1.40e+0 (9.04e-2) |
| Laplace | 6.42e+0 (3.42e-1) | **5.70e-2 (8.16e-3)** | 7.23e-1 (1.39e-1) | 1.64e+0 (4.98e-2) | 1.72e+0 (1.06e-2) | 1.72e+0 (1.06e-2) |
| **Yacht** | | | | | | |
| Uniform | 6.72e+0 (5.08e-1) | **8.56e-1 (4.99e-2)** | 1.62e+1 (1.81e-1) | 1.62e+1 (2.08e-1) | 1.61e+1 (2.04e-1) | 1.61e+1 (1.94e-1) |
| Constant | 3.50e+1 (1.61e+0) | **8.27e-1 (5.78e-2)** | 1.60e+1 (2.11e-1) | 1.59e+1 (2.02e-1) | 1.55e+1 (1.73e-1) | 1.58e+1 (1.95e-1) |
| Student-$t$ | 2.11e+1 (3.51e+0) | **1.12e+0 (2.03e-1)** | 1.64e+1 (2.17e-1) | 1.67e+1 (2.15e-1) | 1.64e+1 (2.08e-1) | 1.66e+1 (2.02e-1) |
| Laplace | 1.61e+1 (1.97e+0) | **1.15e+0 (1.84e-1)** | 1.63e+1 (2.10e-1) | 1.64e+1 (4.06e-1) | 1.63e+1 (2.05e-1) | 1.66e+1 (1.99e-1) |
| **CA Housing** | | | | | | |
| Uniform | **7.10e-1 (7.58e-3)** | 7.39e-1 (1.75e-2) | 1.16e+0 (1.79e-3) | 1.17e+0 (3.04e-3) | 1.17e+0 (2.93e-3) | 1.18e+0 (6.17e-3) |
| Constant | 2.28e+0 (5.12e-2) | **6.35e-1 (4.38e-3)** | 1.17e+0 (2.39e-3) | 1.16e+0 (1.87e-3) | 1.16e+0 (1.88e-3) | 1.17e+0 (5.51e-3) |
| Student-$t$ | 1.34e+0 (1.68e-1) | **6.56e-1 (5.46e-3)** | 1.18e+0 (3.56e-3) | 1.19e+0 (4.13e-3) | 1.18e+0 (3.83e-3) | 1.19e+0 (6.74e-3) |
| Laplace | 1.00e+0 (5.04e-2) | **6.51e-1 (4.71e-3)** | 1.18e+0 (3.41e-3) | 1.18e+0 (3.91e-3) | 1.18e+0 (3.67e-3) | 1.18e+0 (6.30e-3) |

Table 5: Comparison with Huber GP: Negative log predictive density with 15% Corruptions

| Data | Standard | Relevance Pursuit | Student-$t$ | Laplace | Huber | Huber + Projection |
|---|---|---|---|---|---|---|
| **Neal** | | | | | | |
| Uniform | 6.87e+0 (4.15e+0) | **3.80e+0 (4.01e+0)** | **1.85e+0 (9.93e-2)** | **1.64e+0 (1.20e-1)** | **1.70e+0 (1.23e-1)** | **1.70e+0 (1.23e-1)** |
| Constant | 1.32e+2 (1.29e+2) | **-2.50e+0 (2.99e-1)** | 2.03e+0 (1.84e-1) | 8.67e-1 (4.32e-2) | 8.84e-1 (3.87e-2) | 8.84e-1 (3.87e-2) |
| Student-$t$ | **1.98e+0 (9.67e-1)** | **1.99e-1 (2.29e+0)** | **1.67e+0 (1.17e-1)** | **1.25e+0 (8.32e-2)** | **1.28e+0 (8.03e-2)** | **1.28e+0 (8.03e-2)** |
| Laplace | 3.07e+2 (2.15e+2) | **2.02e+0 (2.83e+0)** | 1.73e+0 (1.10e-1) | **1.20e+0 (8.89e-2)** | **1.21e+0 (8.99e-2)** | **1.21e+0 (8.99e-2)** |
| **Friedman 5** | | | | | | |
| Uniform | 2.07e+0 (4.88e-2) | **-1.35e+0 (7.22e-2)** | 4.67e-1 (3.00e-1) | 8.88e-1 (1.01e-1) | 8.29e-1 (9.14e-2) | 8.88e-1 (1.08e-1) |
| Constant | 4.28e+0 (2.25e-2) | **-1.13e+0 (1.82e-1)** | 8.24e-1 (3.17e-1) | 2.28e+0 (2.63e-2) | 2.34e+0 (3.13e-2) | 2.34e+0 (3.13e-2) |
| Student-$t$ | 3.42e+0 (1.34e-1) | **-1.11e+0 (1.04e-1)** | 1.78e-2 (2.86e-1) | 1.55e+0 (1.14e-1) | 1.66e+0 (1.11e-1) | 1.59e+0 (1.10e-1) |
| Laplace | 3.55e+0 (7.84e-2) | **-1.22e+0 (5.89e-2)** | 3.47e-1 (3.13e-1) | 1.75e+0 (8.46e-2) | 1.90e+0 (7.36e-2) | 1.90e+0 (7.36e-2) |
| **Friedman 10** | | | | | | |
| Uniform | 1.86e+0 (3.04e-2) | **-1.82e+0 (3.97e-2)** | 6.78e-2 (3.85e-1) | 9.20e-1 (1.99e-1) | 9.39e-1 (2.01e-1) | 7.65e-1 (1.80e-1) |
| Constant | 4.21e+0 (1.39e-2) | **-1.27e+0 (1.10e-1)** | 8.85e-1 (3.79e-1) | 2.20e+0 (1.45e-2) | 2.23e+0 (1.54e-2) | 2.23e+0 (1.54e-2) |
| Student-$t$ | 3.34e+0 (1.24e-1) | **-1.40e+0 (1.17e-1)** | -6.80e-2 (3.65e-1) | 1.96e+0 (1.25e-1) | 1.95e+0 (1.23e-1) | 1.96e+0 (1.22e-1) |
| Laplace | 3.46e+0 (6.68e-2) | **-1.44e+0 (5.91e-2)** | 4.89e-1 (3.94e-1) | 2.13e+0 (6.35e-2) | 2.21e+0 (2.01e-2) | 2.21e+0 (2.01e-2) |
| **Yacht** | | | | | | |
| Uniform | 3.45e+0 (1.42e-1) | **2.37e+0 (2.79e-1)** | 1.18e+2 (1.06e+1) | 7.07e+1 (5.17e+0) | 7.46e+1 (5.38e+0) | 1.89e+2 (1.64e+1) |
| Constant | 5.29e+0 (1.70e-1) | **1.79e+0 (2.43e-1)** | 7.16e+1 (7.68e+0) | 2.32e+1 (1.59e+0) | 1.95e+1 (1.12e+0) | 3.89e+1 (3.46e+0) |
| Student-$t$ | 4.46e+0 (3.08e-1) | **2.42e+0 (3.38e-1)** | 1.32e+2 (1.05e+1) | 8.20e+1 (5.07e+0) | 8.13e+1 (5.17e+0) | 1.55e+2 (1.38e+1) |
| Laplace | 4.35e+0 (2.88e-1) | **2.27e+0 (3.71e-1)** | 1.14e+2 (9.86e+0) | 6.85e+1 (4.72e+0) | 6.12e+1 (3.87e+0) | 1.17e+2 (8.89e+0) |
| **CA Housing** | | | | | | |
| Uniform | **3.14e+0 (1.44e-1)** | **2.92e+0 (2.06e-1)** | 5.85e+1 (9.59e-1) | 6.77e+1 (2.41e+0) | 6.69e+1 (1.94e+0) | 8.94e+1 (2.45e+1) |
| Constant | **3.57e+0 (2.26e-1)** | 4.51e+0 (2.11e-1) | 4.02e+1 (1.59e+0) | 1.83e+1 (6.58e-1) | 1.75e+1 (7.03e-1) | 2.18e+1 (4.19e+0) |
| Student-$t$ | **1.77e+0 (6.65e-2)** | 3.15e+0 (1.67e-1) | 5.95e+1 (1.44e+0) | 5.20e+1 (2.19e+0) | 4.95e+1 (1.75e+0) | 6.90e+1 (2.02e+1) |
| Laplace | **1.61e+0 (4.21e-2)** | 3.51e+0 (1.93e-1) | 5.60e+1 (1.58e+0) | 4.23e+1 (1.72e+0) | 4.06e+1 (1.47e+0) | 5.62e+1 (1.66e+1) |

Table 6: Comparison with Huber GP: Root mean squared error for 100% Laplace noise

| Data | Standard | Relevance Pursuit | Student-$t$ | Laplace | Huber | Huber + Projection |
|---|---|---|---|---|---|---|
| **Neal** | | | | | | |
| Student-$t$ | 1.96e+0 (3.64e-1) | 1.56e+0 (2.48e-1) | **7.39e-1 (4.08e-2)** | 7.54e-1 (3.17e-2) | **7.48e-1 (3.00e-2)** | **7.48e-1 (3.01e-2)** |
| Laplace | 1.51e+0 (1.06e-1) | 2.40e+0 (2.72e-1) | **1.16e+0 (9.18e-2)** | **1.12e+0 (9.90e-2)** | **1.12e+0 (9.80e-2)** | **1.12e+0 (9.80e-2)** |
| **Friedman 5** | | | | | | |
| Student-$t$ | 1.67e+1 (1.83e+0) | 9.30e+0 (3.05e-1) | 5.26e+0 (1.30e-1) | **4.57e+0 (1.60e-1)** | **4.61e+0 (1.55e-1)** | **4.61e+0 (1.55e-1)** |
| Laplace | 1.39e+1 (5.95e-1) | 1.37e+1 (6.36e-1) | 8.33e+0 (3.69e-1) | **7.34e+0 (3.50e-1)** | **7.40e+0 (3.45e-1)** | **7.40e+0 (3.45e-1)** |
| **Friedman 10** | | | | | | |
| Student-$t$ | 2.10e+1 (3.42e+0) | 7.80e+0 (2.03e-1) | 4.72e+0 (1.14e-1) | **4.02e+0 (9.57e-2)** | **4.05e+0 (9.34e-2)** | **4.05e+0 (9.34e-2)** |
| Laplace | 1.30e+1 (3.77e-1) | 1.27e+1 (3.99e-1) | 7.24e+0 (1.99e-1) | **6.07e+0 (1.91e-1)** | **6.26e+0 (1.71e-1)** | **6.26e+0 (1.71e-1)** |
| **Yacht** | | | | | | |
| Student-$t$ | 5.89e+1 (1.37e+1) | 2.44e+1 (1.77e+0) | **1.57e+1 (1.36e-1)** | **1.57e+1 (1.51e-1)** | **1.57e+1 (1.51e-1)** | 1.60e+1 (3.16e-1) |
| Laplace | 2.57e+1 (1.17e+0) | 4.75e+1 (3.51e+0) | **1.64e+1 (3.18e-1)** | **1.61e+1 (2.52e-1)** | **1.62e+1 (2.60e-1)** | 1.67e+1 (3.55e-1) |
| **CA Housing** | | | | | | |
| Student-$t$ | 2.92e+0 (5.80e-1) | **1.24e+0 (6.17e-2)** | **1.17e+0 (3.65e-3)** | 1.18e+0 (4.42e-3) | **1.17e+0 (4.14e-3)** | 1.21e+0 (3.05e-2) |
| Laplace | 1.57e+0 (8.25e-2) | 2.06e+0 (1.44e-1) | **1.21e+0 (1.08e-2)** | **1.20e+0 (9.39e-3)** | **1.20e+0 (9.69e-3)** | 1.23e+0 (2.05e-2) |

Table 7: Comparison with Huber GP: Negative log predictive density with 100% Laplace noise

| Data | Standard | Relevance Pursuit | Student-$t$ | Laplace | Huber | Huber + Projection |
|---|---|---|---|---|---|---|
| Neal | | | | | | |
| Student-$t$ | 1.95e+0 (1.22e-1) | 8.78e+1 (2.80e+1) | **1.17e+0 (6.08e-2)** | **1.22e+0 (5.71e-2)** | **1.20e+0 (5.17e-2)** | **1.21e+0 (5.15e-2)** |
| Laplace | 1.89e+0 (1.14e-1) | 1.38e+2 (3.98e+1) | **1.57e+0 (9.22e-2)** | **1.55e+0 (9.20e-2)** | **1.55e+0 (9.12e-2)** | **1.55e+0 (9.12e-2)** |
| Friedman 5 | | | | | | |
| Student-$t$ | 4.43e+0 (8.19e-2) | 3.93e+0 (2.19e-2) | 3.11e+0 (2.69e-2) | **2.96e+0 (3.49e-2)** | **2.97e+0 (3.27e-2)** | **2.97e+0 (3.27e-2)** |
| Laplace | 4.55e+0 (2.81e-2) | 4.56e+0 (1.88e-2) | 3.64e+0 (4.03e-2) | **3.46e+0 (4.45e-2)** | **3.48e+0 (4.31e-2)** | **3.48e+0 (4.31e-2)** |
| Friedman 10 | | | | | | |
| Student-$t$ | 4.60e+0 (8.54e-2) | 3.89e+0 (1.36e-2) | 3.04e+0 (2.45e-2) | **2.83e+0 (2.30e-2)** | **2.84e+0 (2.22e-2)** | **2.84e+0 (2.22e-2)** |
| Laplace | 4.58e+0 (1.25e-2) | 4.57e+0 (1.27e-2) | 3.60e+0 (2.53e-2) | **3.32e+0 (3.01e-2)** | **3.37e+0 (2.61e-2)** | **3.37e+0 (2.61e-2)** |
| Yacht | | | | | | |
| Student-$t$ | 5.14e+0 (1.85e-1) | **4.60e+0 (8.15e-2)** | 7.30e+0 (1.40e-1) | 6.79e+0 (1.52e-1) | 6.70e+0 (1.32e-1) | 1.28e+1 (7.75e-1) |
| Laplace | **4.74e+0 (4.45e-2)** | 5.30e+0 (8.31e-2) | 4.91e+0 (8.87e-2) | 5.19e+0 (1.17e-1) | 5.12e+0 (1.04e-1) | 9.90e+0 (6.44e-1) |
| CA Housing | | | | | | |
| Student-$t$ | 2.08e+0 (1.01e-1) | **1.79e+0 (6.40e-2)** | 7.41e+0 (2.39e-1) | 6.52e+0 (2.54e-1) | 6.41e+0 (2.07e-1) | 6.83e+0 (3.28e-1) |
| Laplace | **1.88e+0 (5.25e-2)** | 2.24e+0 (9.41e-2) | 2.92e+0 (6.01e-2) | 3.33e+0 (7.47e-2) | 3.27e+0 (5.84e-2) | 6.64e+0 (3.28e+0) |

