# OpenReview forum: "Robust Gaussian Processes via Relevance Pursuit"
_NeurIPS.cc/2024/Conference — NeurIPS 2024 poster_

### Official Review · Reviewer_geQY · 2024-07-02

**Soundness:** 3
**Presentation:** 3
**Contribution:** 3
**Rating:** 7
**Confidence:** 4

**Summary:**

This work proposes a new way to perform heteroskedastic regression using Gaussian processes via data-point-specific noise levels. These noise levels are inferred using a sequential selection procedure maximizing the log-marginal likelihood. The authors show that under a specific parametrization the log-marginal likelihood is strongly concave in the noise variances and give approximation guarantees for their proposed algorithm. Experiments on regression and Bayesian optimization tasks demonstrate the benefits of this approach.

**Strengths:**

Making Gaussian processes robust to some degree of model misspecification is a worthwhile goal and this paper provides a nice addition to the existing toolbox of methods. The paper is well-written and presents largely well-executed experiments for regression and Bayesian optimization. In particular, in tasks where Bayesian optimization is applied, it seems plausible that outliers appear in the way they are synthetically generated in the evaluation. I also appreciated that the authors submitted their code.

**Weaknesses:**

### Theory
While correct as far as I can tell, I question the value of the theoretical result, if in practice you optimize the hyperparameters of the covariance function and $\rho$ jointly. Is there reason to believe, that the convexity in $\rho$ for fixed hyperparameters is beneficial given this choice? It would be informative to see an experiment that compares the non-convex parametrization to the convex one where all hyperparameters are optimized jointly. If the convex parametrization improves convergence, this would suggest that convexity is beneficial even when optimizing all hyperparameters jointly.

### Related Work
Even though this is quite recent work, I would have liked to see a discussion of Altamirano et al. (2024) in the related work section. They introduce data-point-specific observation noise based on a generalized Bayesian inference perspective. I think a comparison to their approach in the experiments would improve the paper, but I would not expect this given the recency of the work.

- Altamirano, Matias, Francois-Xavier Briol, and Jeremias Knoblauch. "Robust and Conjugate Gaussian Process Regression." International Conference on Machine Learning (ICML), 2024. URL: https://arxiv.org/abs/2311.00463

### Experiments
As the authors laudably acknowledge, the fit times were 100x slower (see Figure 8) than the simple baseline approaches. How does that impact the Bayesian optimization results if plotted as a function of wall-clock time? It seems that eventually some of the other approaches catch up to your method. If the fit time is 100x that of the baseline approaches, the reward curves might look different if plotted as a function of wall-clock time. Or is there an implicit assumption that each evaluation is significantly more expensive than fitting the GP?
The experiments on regression problems could be expanded in my opinion, e.g. with benchmark datasets from the UCI repository. The test problems that were considered are arguably designed to test optimization algorithms (hence their usage in BO).
From looking at the code the "real-world problems" in Section 6 have synthetically generated corruptions. I think the paper could be made stronger if there were at least one experiment where the corruption is not synthetically generated, but simply part of the observation process.

### Other Weaknesses and Improvements
- Almost all plots are way too small and barely readable. If you reformat these, then I am willing to raise my "Presentation" score.
- Link to the proofs in the appendix after each theoretical result, otherwise they are hard to find.
- When you reference details in the appendix, link to the appropriate section (e.g. line 286).
- Citations in lines 32 to 34 for the methods that are referenced are missing
- Typos (l.211, l. 286, l. 362)

**Questions:**

- Is your assumption about the data corruption different from Huber's $\varepsilon$-contamination model (see e.g. https://arxiv.org/pdf/1511.04144) or is it identical? Equation 2 suggests they are different, but the text and experiments often refer to only a fraction of the data being corrupted (e.g. in Section 4.3 and Section 6).
- How does your approach compare to a vanilla GP on a dataset with no corruption? Does it "fail gracefully"?
- Do you see better performance when *jointly* optimizing the hyperparameters for the convex reparametrization of the data-point-specific noise variances than if not reparametrizing?
- How did you set the schedule to test data points for being outliers in the experiments (c.f. lines 182-184)?

**Limitations:**

I would have liked to see a bit more discussion on the limitations of the approach in Section 7. For example, the fact that the theoretical result does not apply when jointly optimizing the hyperparameters as is the case in the experiments, and a discussion of the wall-clock runtime.

---

> ### Author Rebuttal · Authors · 2024-08-07
>
> > “While correct as far as I can tell, I question the value of the theoretical result, if in practice you optimize the hyperparameters of the covariance function and ρ jointly. Is there reason to believe, that the convexity in ρ for fixed hyperparameters is beneficial given this choice?” “It would be informative to see an experiment that compares the non-convex parametrization to the convex one where all hyperparameters are optimized jointly”
>
> We would expect the overall problem to be “more convex” than with the canonical parameterization, and thus improve optimization performance. More formally, the convexity guarantee will guarantee the positive-definiteness of the submatrix of the Hessian corresponding to the rhos, which is better than not guaranteeing positive definiteness at all.
>
> Positive-definiteness is also beneficial for quasi-Newton optimization algorithms like L-BFGS, which restarts the Hessian-approximation whenever it encounters non-convex regions, because the associated updates to the Hessian approximation are not positive-definite. This leads the algorithm to momentarily revert back to gradient descent, with an associated slower convergence rate.
>
> Practically, we ran convergence analyses using the data from Figure 1, allowing all rhos to be optimized jointly with other hyper-parameters (length scale, kernel variance and noise variance), recording the achieved negative log marginal likelihood (NLML) as a function of the tolerance parameter `ftol` of the L-BFGS optimizer. The results indicate that the optimizer terminates with a much better _NLML_ using the convex parameterization with the same convergence tolerance:
>
> ftol | Canonical | Convex |
> ----|----|--------|
> 1e-03 | -4.37 | -14.18 |
> 1e-04 | -4.37 | -93.00 |
> 1e-05 | -4.37 | -93.01 |
> 1e-06 | -4.37 | -135.05 |
> 1e-07 | -98.62 | -518.68 |
> 1e-08 | -97.52 | -1139.29 |
>
> There are also settings in which we do not actually jointly optimize the hyperparameters. For instance, consider a situation in which we have access to data from the same data generating process that has been manually labeled by domain experts as outlier-free. Then we can estimate the model hyperparameters (of the non-robust GP) on that data, and fix those for the RGP-RP on the new data set that we do not know to be outlier-free (and that we cannot label due to cost or other practical reasons).
>
> > “Even though this is quite recent work, I would have liked to see a discussion of Altamirano et al. (2024) in the related work section”
>
> Thanks for pointing out this relevant work. We have added a discussion of and comparison to this work in our general response, and will add this to the paper.
>
> > “Or is there an implicit assumption that each evaluation is significantly more expensive than fitting the GP?”
>
> This is indeed the case; GPs are commonly applied to Bayesian optimization or active learning tasks where evaluation times are on the order of hours or days, so the fitting time of the GP is generally not much of a concern. We will make this assumption explicit.
>
> > “The experiments on regression problems could be expanded in my opinion, e.g. with benchmark datasets from the UCI repository”
>
> We have done so, see our general response. Please let us know if there are any additional datasets you would like us to consider for the CR if accepted.
>
> > “I think the paper could be made stronger if there were at least one experiment where the corruption is not synthetically generated [...]”
>
> Thanks for this feedback. We have added the “Twitter Flash Crash” example from Altamirano et al. (2024) and an additional variant of it to our results (see general response). If you have any other examples of real-world data we would be happy to consider this for inclusion in the CR.
>
> > “Almost all plots are way too small and barely readable”
>
> We have adjusted the size of our plots (in particular the axis labels and legends), see attached pdf. We will utilize part of the additional space for the CR submission to increase plot size throughout.
>
> > “Is your assumption about the data corruption different from Huber's ε-contamination model or is it identical? Equation 2 suggests they are different, but the text and experiments often refer to only a fraction of the data being corrupted [...]”
>
> A significant high-level difference is that Huber’s model is homoskedastic while ours is not. Citing Huber’s original article, he states: “Let $x_1,⋯, x_n$ be independent random variables with common distribution function $F$”, i.e. Huber’s model assumes that every data point is equally likely to be an outlier.
>
> In contrast, our model introduces data-point specific variances that are adapted using marginal likelihood optimization, leading the resulting likelihood function to be different for each datapoint with a distinct rho.
>
> The article often refers to the “fraction of data being corrupted” because the inference of this fraction is an important problem that determines the performance of RP, which requires choosing a number of outlier variances ($\rho_i$). We solve this problem by leveraging Bayesian model selection.
>
> > “How does your approach compare to a vanilla GP on a dataset with no corruption? Does it "fail gracefully”?”
>
> Yes, and the results in our general response demonstrate this. We expect this to hold as long as the prior on the occurrence probability is sufficiently uninformative and puts some weight on no occurrences. Throughout our experiments, we used a geometric prior on the support size with a mean equal to 20% of the number of observations. We will add the results and discussion to the manuscript.
>
> > “How did you set the schedule to test data points for being outliers in the experiments (c.f. lines 182-184)?”
>
> See [our response to reviewer aaaZ](https://openreview.net/forum?id=5FATPIlWUJ&noteId=OjPuKthMyu).
>
> > “I would have liked to see a bit more discussion on the limitations of the approach in Section 7 [...]”
>
> Good point, we will include these in the discussion in Section 7.

---

> > ### Comment · Reviewer_geQY · 2024-08-07
> >
> > Thank you for your response. I believe the additional results you presented strengthen the paper. I have raised my score to reflect this.

---

> > > ### Author Response · Authors · 2024-08-07
> > >
> > > Thank you again for your detailed, actionable feedback, and consideration.

---

### Official Review · Reviewer_aaaZ · 2024-07-09

**Soundness:** 3
**Presentation:** 3
**Contribution:** 3
**Rating:** 7
**Confidence:** 4

**Summary:**

The paper proposes a robust Gaussian Process regression by inferring data-point-specific noise levels with a sequential selection procedure maximising the log marginal likelihood. The authors show the good performance of their method in a mix of synthetic and real-world regression tasks, including Bayesian optimisation.

**Strengths:**

The paper is well-written and easy to follow. The method is clearly motivated and presented, with a good amount of related work discussed. The authors address a problem—robust regression in the context of Gaussian processes—of sufficiently broad interest to merit publication. The authors also provide a theoretical analysis to support their method.

**Weaknesses:**

In my opinion, the weakest part of the paper is the regression problems in the experiment section. It only benchmarks against two test functions, one of which is not a standard benchmark for regression; I would like to see more standard UCI benchmarks. Additionally, the proposed method does not clearly outperform other methods, particularly in the Hartmann6 test, where adaptive Winsorization outperforms RGP-RP. The authors do not discuss the results of the regression problems, so a discussion providing guidelines on the cases where the proposed method will perform better is needed.

Finally, the authors claim their method "permits fast and robust inference"; however, in Appendix D.1, the method is slower than the Student-t GP for the Hartmann6. It would be useful if the authors discussed this in the main paper.

**Questions:**

- What's the difference between RGP-RP and RGP-RP* in the experiment section?
- What's the schedule $\mathcal{K}$ selected for the experiments? How does this affect the performance and fitting time of the method?
- While the Friedman and Hartmann6 test functions are well-known, the authors should provide a reference for these functions.
- Why use Hartmann6 as a test function for regression? Hartmann6 is typically used to test optimisation due to its properties. Why not use standard UCI benchmarks?
- Do the authors know why the student-t GP performs so poorly in the Hartmann6 function, even when the noise comes from a student-t distribution?

**Limitations:**

No obvious negative societal impact

---

> ### Author Rebuttal · Authors · 2024-08-07
>
> > “The weakest part of the paper is the regression problems in the experiment section”
>
> As part of our rebuttal, we have produced a significant number of additional results on UCI benchmarks, the benchmarks and real-world Twitter crash example of Altamirano et al. 2024, and included an additional baseline (see general response and attached pdf).
>
> > “Additionally, the proposed method does not clearly outperform other methods, particularly in the Hartmann6 test, where adaptive Winsorization outperforms RGP-RP.”
>
> We don’t necessarily expect our method to outperform all other baselines in all settings. But with our additional empirical results in our general response, we do show that it (i) works much better in situations in which outliers cannot be easily separated from the rest of the data, (ii) is competitive in most other settings. We also note that while the adaptive winsorization heuristic indeed performs slightly better than RGP-RP on Hartmann6 with constant, well-separated outliers, it completely fails on the Friedmann10 function and is barely better than a non-robust GP. In general, winsorization does not perform well if the outliers are not clearly and a-priori separated from the rest of the data.
>
> > “The authors do not discuss the results of the regression problems, so a discussion providing guidelines on the cases where the proposed method will perform better is needed”
>
> We agree that the discussion can be improved and we will include this into the paper (see general response for discussion).
>
> > “the authors claim their method "permits fast and robust inference"; however, in Appendix D.1, the method is slower than the Student-t GP for the Hartmann6. It would be useful if the authors discussed this in the main paper”
>
> The Student-t likelihood performs very poorly on Hartmann6 and has very high variance. We suspect that there are numerical issues in the optimization of the variational objective that results in the fitting terminating early and at suboptimal points, thus producing faster fit times. Note that the Student-t log likelihood is also non-convex, which can give rise to highly suboptimal local minima. We plan to investigate this in more detail.
>
> > “What's the difference between RGP-RP and RGP-RP* in the experiment section?”
>
> See [our response to reviewer CPiv](https://openreview.net/forum?id=5FATPIlWUJ&noteId=JGfjtKQW5T).
>
> > “What's the schedule K selected for the experiments? How does this affect the performance and fitting time of the method?”
>
> The default schedule used throughout the experiments is `[0%, 5%, 10%, 15%, 20%, 30%, 40%, 50%, 75%, 100%]` of the number of observations (traversed backwards by the backward algorithm). The fitting time is linear in the number of steps in the schedule, as a new model fit is carried out for each step of the schedule.
>
> The advantage of a schedule with a finer granularity is that the model could be more data efficient, by introducing additional noise variances for outlying data points only, so that all other data is "trusted" up to the homoskedastic noise variance $\sigma^2$. For most practical applications, it is probably sufficient to test for single-digit percentages of outliers, the default schedule here is designed to exhibit high generality, as is evidenced by RP's performance on the variety of benchmarks, including its outperformance on the RCGP benchmarks and Twitter crash example of Altamirano et al. 2024 included in the general response, using the same schedule.
>
> If timing was a particular concern, the optimization could likely be accelerated substantially by keeping track of the approximation of the Hessian matrix generated by each L-BFGS call, which could be used to warm-start the optimization of each step and lead to significantly accelerated convergence, as the consecutive optimization problems tend to be similar.
>
> > “While the Friedman and Hartmann6 test functions are well-known, the authors should provide a reference for these functions.”
>
> Thanks for the callout, we will provide the respective references.
>
> > “Why use Hartmann6 as a test function for regression? Hartmann6 is typically used to test optimisation due to its properties. Why not use standard UCI benchmarks?”
>
> We have produced additional results on a number of UCI benchmarks as part of this rebuttal (see general response).
>
> > “Do the authors know why the student-t GP performs so poorly in the Hartmann6 function, even when the noise comes from a student-t distribution?”
>
> We believe you are referring to the right column of Figure 4, whose x-axis corresponds to the fraction of data points that are corrupted by the outlier process. First, the axis label is missing in the submission, and we will correct this.
>
> What the figure shows is that the Student-t GP only starts to dominate the other methods once more than 40%-50% of data points are corrupted by Student-t noise. We indeed expect the Student-t GP to be the best model as the fraction of Student-t-distributed noise increases, as it becomes the correct noise assumption when 100% of data points are “corrupted” by Student-t noise. However, the results also show: if the main goal is to make a GP robust to a _sparse_ set of outliers – rather than a uniformly heavy tailed observation noise – then RP significantly outperforms the Student-t GP by leveraging this sparse structure.

---

> > ### Comment · Reviewer_aaaZ · 2024-08-12
> >
> > Thank you for your response. I'll increase my score to 7.

---

### Official Review · Reviewer_CPiv · 2024-07-12

**Soundness:** 2
**Presentation:** 3
**Contribution:** 2
**Rating:** 7
**Confidence:** 5

**Summary:**

The authors propose a robust Gaussian process model in the sparse heteroscedastic noise setting. Orthogonal Matching Pursuit- like algorithm is proposed to infer data point specific noise levels. The negative log marginal likelihood function to be optimized is claimed to be strongly convex by reparametrizing the additive noise levels. The method is validated on synthetic as well as real-world experiments.

**Strengths:**

1. The authors formulated a convex $\mathcal{L}$ in the Gaussian process regression model by reparametrizing the noise levels, which is very desirable for optimization perspective. The representation of the paper is good, however, I found the section 4.3 difficult to follow.
2. The relevance pursuit is a novel approach to the heteroscedastic noise case of robust GP regression.

**Weaknesses:**

1. Much more work on heavy tailed outliers is done. So, more references and discussion need to be added in related work. Kersting et. al. 2007 proposed GPs for heteroscedastic noise setting. They also learned the homoscedastic part of the variance aiming to learn the remaining part using a second GP. The work is related and at least needs a mention. It would have been nice to see comparison with this model.

2. While this work employs greedy algorithms for GPs, it is not the first to do so, so I would not consider it exceptionally novel.

3. The experimentation is adequate but less discussed.

Ref: Most Likely Heteroscedastic Gaussian Process Regression


Minor corrections:

1. Consider adding references for these three categories on line 34. No reference is given for down weighting procedures even in related work.
2. Figure 1 and 2 are not discussed at all. What regression example?
3. broken line 150
4. unclosed braces figure 3
5. no need of repeating the abbreviation RIP on line 200
6. Broken line 264
7. Is the x axis of right figure in Figure 4 percentage of corruptions or probability?
8. Different notations need to be used for $\delta$ on lines 67 and 112.

**Questions:**

1. Why Matern kernel was chosen in one of the baselines?
2. What is the difference between RGP-RP*-BW and RGP-RP-BW. I see that the former is canonical one. What is the difference between canonical and non-canonical reparameterization of $\mathbf{s}$?
3. What is $\mathbf{K}_{0}$ on line 218?

**Limitations:**

Limitations are discussed as I hoped they would be.

---

> ### Author Rebuttal · Authors · 2024-08-07
>
> > Kersting et. al. 2007:
>
> This is indeed related work, and we will discuss it in the updated manuscript. The main difference to our method is that their approach is unlikely to work well with outliers. In particular, while they do account for heteroskedastic observation noise, they model the logarithm of the standard deviation of the noise noise process as another GP. This not just restricts the observation to be (heteroskedastic) Gaussian, but, importantly, imposes a smoothness assumption on the variance of the observations as a function of the inputs. In contrast, our approach infers individual additional noise terms, and thus allows us to go beyond the setting in Kersting et. al. 2007 to handle even gross outliers (which require a noise process that is discontinuous in the inputs). Another downside of the approach in Kersting et. al. 2007 is that they require an EM algorithm to estimate the parameters of the two GP models, which does not have any theoretical guarantees and in practice often has convergence issues (see e.g. discussion here).
>
> > “I found the section 4.3 difficult to follow”
>
> Thank you for your feedback. We will improve the clarity of our writing in this section in particular by introducing the general framework of Bayesian model selection before stating its instantiation in our work and providing further background references. Please let us know if there are additional edits you would like us to do.
>
> > “While this work employs greedy algorithms for GPs, it is not the first to do so”
>
> While greedy algorithms have been applied for many problems in general (outside of GPs), their application to the sparse optimization of noise variances for robustness, which we show permits strong theoretical guarantees, has to our knowledge not been studied and proposed before. Please let us know if you would like us to add any additional reference for related work.
>
> > “The experimentation is adequate but less discussed”
>
> We produced a significant number of additional results with new baselines and test problems for this rebuttal, and will improve discussion of the entirety of the empirical results (see general response).
>
> > “Figure 1 and 2 are not discussed at all. What regression example?”
>
> Thank you for catching this. The illustrative regression example is based on a synthetic one-dimensional modified sine function with a few outliers. We will detail the exact setup for this in the appendix.
>
> > “Is the x axis of right figure in Figure 4 percentage of corruptions or probability?”
>
> Yes, this is stated in the figure’s caption. We will add an axis label as well in order to improve clarity.
>
> > "Why Matern kernel was chosen in one of the baselines?”
>
> No particular reason other than that the Matern-5/2 kernel is widely used in practice and the default in many GP / BO libraries (e.g. botorch, trieste). We expect the results to be quite similar if another common kernel, say an RBF kernel, were used.
>
> > “What is the difference between RGP-RP*-BW and RGP-RP-BW. I see that the former is canonical one. What is the difference between canonical and non-canonical reparameterization of s?”
>
> This difference is described at the beginning of Section 5.2. The “canonical” parameterization is that which directly uses the robust variances $\boldsymbol \rho$ in the parameterization. The non-canonical parameterization is the convex parameterization we introduce in Section 5.2, i.e., that where ${\boldsymbol \rho}(\mathbf s) = \text{diag}({\bf K_0}) \odot ((1 - {\mathbf s})^{-1} - 1)$ and the new parameter is $\mathbf{s}$. The method label coding is as follows: A “*” indicates using the canonical (not necessarily convex) parameterization, and “BW” indicates using the backward algorithm for relevance pursuit (Algorithm 2 in Appendix A). We will improve the clarity of the exposition here and clearly explain the method labels. The convex parameterization is a key ingredient that allows us to prove strong theoretical guarantees for the result of the greedy algorithm, and also has beneficial effects on the convergence of the hyper-parameter optimization algorithms, as we demonstrate in the other responses.
>
> > “What is $\bf K_0$ on line 218?”
>
> Thanks for catching this forward reference, $\mathbf{K}_0$ is defined in Theorem 8: $\mathbf{K}_0 = k(\mathbf{X}, \mathbf{X}) + \sigma^2 \bf I$. We’ll move the definition up in the text.

---

> > ### Comment · Reviewer_CPiv · 2024-08-11
> >
> > Thank you for addressing my concerns in detail. While the additional experimentation certainly demonstrates the effectiveness of the RGP, a more comprehensive comparison would benefit from including benchmarks that explicitly model heavy-tailed distributions, such as those employing Laplace (Kuss et. al., 2006) and Huber (Algikar et.al., 2023) likelihoods. Both methods address similar challenges – spatially uncorrelated gross outliers and input-independent noise – making them valuable benchmarks. Algikar et.al's method claims to not lose efficiency at Gaussian distribution using Huber loss and leverage weighting. My current score remains unchanged.

---

> > > ### Author Response · Authors · 2024-08-12
> > > **[time sensitive question]**
> > >
> > > We are glad we addressed your concerns in detail via our additional experiments which demonstrate the effectiveness of RGP.
> > >
> > > The reviewer now suggests additional methods to compare against. At a minimum, we can discuss our work in the context of those papers, and promise to include these comparisons in the CR.   We hope that time permits us to run some additional simulations in the next day to meet your standards.
> > >
> > > Before we can do this, can you please clarify the stated references in your follow-up response above?
> > >
> > > Is "Kuss et. al 2006" Malte Kuß's PhD thesis https://pure.mpg.de/rest/items/item_1791134/component/file_3167621/content ?
> > >
> > > Is "Algikar et.al., 2023" the unpublished manuscript of Pooja Algikar & Lamine Mili, Robust Gaussian Process Regression with Huber Likelihood?
> > > (preprint at https://www.researchgate.net/publication/367280832_Robust_Gaussian_Process_Regression_with_Huber_Likelihood)

---

> > > > ### Comment · Reviewer_CPiv · 2024-08-12
> > > >
> > > > I don't intend to create last minute pressure of additional experimentation on you on the last day. I am glad to see that you offer the comparison with the requested models. I am okay with you adding them in the final submission of the paper. You got the references for the preprints right. I will change my score to 6.

---

> ### Author Response · Authors · 2024-08-13
> **Results for Suggested Benchmarks**
>
> Thank you again for your suggestions. We present the additional results you suggested below and hope that this will allow you to fully recommend our work for acceptance.
>
> **Summary** We added additional variational GP models with Laplace and Huber likelihoods, and translated the Matlab code of the "projection statistics" of Algikar et.al., 2023 ([Github](https://github.com/apooja1/GP-Huber/tree/main)) to PyTorch. We then combined the projection-statistics-based weighting of the Huber loss with a variational GP (referred to as `Huber-Projection` ) to get as close as possible to a direct comparison to Algikar et.al., 2023 without access to a Matlab license.
>
> ## Additional Benchmark Results with 15% Corruptions (Negative Log Likelihood)
>
> The following results include the Friedman 10 function used in Algikar et.al., 2023, and the UCI CA Housing data set, which - according to UCI - is a replacement for the deprecated UCI Boston Housing data, which was used in Algikar et.al., 2023.
>
> In summary, for a sparse 15% of corrupted observations, Relevance Pursuit is able to almost uniformly achieve better predictive negative log likelihoods (NLL) for the datasets and outlier distributions we consider, the only exception being the CA Housing data, where the standard GP performs surprisingly well in terms of NLL, but not in terms of RMSE.
>
> | Data | Standard | Relevance Pursuit | Student-t | Laplace | Huber | Huber + Projection |
> | --- | --- | --- | --- | --- | --- | --- |
> friedman5 + uniform | 2.07e+00 (4.88e-02) | **-1.35e+00 (7.22e-02)** | 4.67e-01 (3.00e-01) | 8.88e-01 (1.01e-01) | 8.29e-01 (9.14e-02) | 8.88e-01 (1.08e-01)
> friedman5 + constant | 4.28e+00 (2.25e-02) | **-1.13e+00 (1.82e-02)** | 8.24e-01 (3.17e-01) | 2.28e+00 (2.63e-02) | 2.34e+00 (3.13e-02) | 2.34e+00 (3.13e-02)
> friedman5 + student-t | 3.42e+00 (1.34e-01) | **-1.11e+00 (1.04e-01)** | 1.78e-02 (2.86e-01) | 1.55e+00 (1.14e-01) | 1.66e+00 (1.11e-01) | 1.59e+00 (1.10e-01)
> friedman5 + laplace | 3.55e+00 (7.84e-02) | **-1.22e+00 (5.89e-02)** | 3.47e-01 (3.13e-01) | 1.75e+00 (8.46e-02) | 1.90e+00 (7.36e-02) | 1.90e+00 (7.36e-02)
> friedman10 + uniform | 1.86e+00 (3.04e-02) | **-1.82e+00 (3.97e-02)** | 6.78e-02 (3.85e-01) | 9.20e-01 (1.99e-01) | 9.39e-01 (2.01e-01) | 7.65e-01 (1.80e-01)
> friedman10 + constant | 4.21e+00 (1.39e-02) | **-1.27e+00 (1.10e-02)** | 8.85e-01 (3.79e-01) | 2.20e+00 (1.45e-02) | 2.23e+00 (1.54e-02) | 2.23e+00 (1.54e-02)
> friedman10 + student-t | 3.34e+00 (1.24e-01) | **-1.40e+00 (1.17e-01)** | -6.80e-02 (3.65e-01) | 1.96e+00 (1.25e-01) | 1.95e+00 (1.23e-01) | 1.96e+00 (1.22e-01)
> friedman10 + laplace | 3.46e+00 (6.68e-02) | **-1.44e+00 (5.91e-02)** | 4.89e-01 (3.94e-01) | 2.13e+00 (6.35e-02) | 2.21e+00 (2.01e-02) | 2.21e+00 (2.01e-02)
> yacht_hydrodynamics + uniform | 3.45e+00 (1.42e-01) | **2.37e+00 (2.79e-01)** | 1.18e+02 (1.06e+01) | 7.07e+01 (5.17e+00) | 7.46e+01 (5.38e+00) | 1.89e+02 (1.64e+01)
> yacht_hydrodynamics + constant | 5.29e+00 (1.70e-01) | **1.79e+00 (2.43e-01)** | 7.16e+01 (7.68e+00) | 2.32e+01 (1.59e+00) | 1.95e+01 (1.12e+00) | 3.89e+01 (3.46e+00)
> yacht_hydrodynamics + student-t | 4.46e+00 (3.08e-01) | **2.42e+00 (3.38e-01)** | 1.32e+02 (1.05e+01) | 8.20e+01 (5.07e+00) | 8.13e+01 (5.17e+00) | 1.55e+02 (1.38e+01)
> yacht_hydrodynamics + laplace | 4.35e+00 (2.88e-01) | **2.27e+00 (3.71e-01)** | 1.14e+02 (9.86e+00) | 6.85e+01 (4.72e+00) | 6.12e+01 (3.87e+00) | 1.17e+02 (8.89e+00)
> california_housing + uniform | **3.14e+00 (1.44e-01)** | **2.92e+00 (2.06e-01)** | 5.85e+01 (9.59e-01) | 6.77e+01 (2.41e+00) | 6.69e+01 (1.94e+00) | 8.94e+01 (2.45e+01)
> california_housing + constant | **3.57e+00 (2.26e-01)** | 4.51e+00 (2.11e-01) | 4.02e+01 (1.59e+00) | 1.83e+01 (6.58e-01) | 1.75e+01 (7.03e-01) | 2.18e+01 (4.19e+00)
> california_housing + student-t | **1.77e+00 (6.65e-02)** | 3.15e+00 (1.67e-01) | 5.95e+01 (1.44e+00) | 5.20e+01 (2.19e+00) | 4.95e+01 (1.75e+00) | 6.90e+01 (2.02e+01)
> california_housing + laplace | **1.61e+00 (4.21e-02)** | 3.51e+00 (1.93e-01) | 5.60e+01 (1.58e+00) | 4.23e+01 (1.72e+00) | 4.06e+01 (1.47e+00) | 5.62e+01 (1.66e+01)
>
> ## RMSE
>
> RP performs uniformly better w.r.t. RMSE, we only include CA Housing here due to space limitations.
>
> | Data | Standard | Relevance Pursuit | Student-t | Laplace | Huber | Huber + Projection |
> | --- | --- | --- | --- | --- | --- | --- |
> california_housing + uniform | **7.10e-01 (7.58e-03)** | 7.39e-01 (1.75e-02) | 1.16e+00 (1.79e-03) | 1.17e+00 (3.04e-03) | 1.17e+00 (2.93e-03) | 1.18e+00 (6.17e-03)
> california_housing + constant | 2.28e+00 (5.12e-02) | **6.35e-01 (4.38e-03)** | 1.17e+00 (2.39e-03) | 1.16e+00 (1.87e-03) | 1.16e+00 (1.88e-03) | 1.17e+00 (5.51e-03)
> california_housing + student-t | 1.34e+00 (1.68e-01) | **6.56e-01 (5.46e-03)** | 1.18e+00 (3.56e-03) | 1.19e+00 (4.13e-03) | 1.18e+00 (3.83e-03) | 1.19e+00 (6.74e-03)
> california_housing + laplace | 1.00e+00 (5.04e-02) | **6.51e-01 (4.71e-03)** | 1.18e+00 (3.41e-03) | 1.18e+00 (3.91e-03) | 1.18e+00 (3.67e-03) | 1.18e+00 (6.30e-03)

---

> ### Author Response · Authors · 2024-08-13
> **Results for uniformly heavier-tailed noise**
>
> **We highlight the main limitation** of Relevance Pursuit versus the additional baselines: if 100% of the data points are subject to heavy-tailed noise, the methods based on uniformly heavy-tailed likelihoods, including `Huber + Projection` will perform best, as the following results demonstrate, similar to the setup in Table 1 of Algikar et.al., 2023.
>
> # 100% Laplace Noise
>
> ## RMSE
>
> | Data | Standard | Relevance Pursuit | Student-t | Laplace | Huber | Huber + Projection |
> | --- | --- | --- | --- | --- | --- | --- |
> neal + laplace | 1.51e+00 (1.06e-01) | 2.40e+00 (2.72e-01) | **1.16e+00 (9.18e-02)** | **1.12e+00 (9.90e-02)** | **1.12e+00 (9.80e-02)** | **1.12e+00 (9.80e-02)**
> friedman5 + laplace | 1.39e+01 (5.95e-01) | 1.37e+01 (6.36e-01) | 8.33e+00 (3.69e-01) | **7.34e+00 (3.50e-01)** | **7.40e+00 (3.45e-01)** | **7.40e+00 (3.45e-01)**
> friedman10 + laplace | 1.30e+01 (3.77e-01) | 1.27e+01 (3.99e-01) | 7.24e+00 (1.99e-01) | **6.07e+00 (1.91e-01)** | **6.26e+00 (1.71e-01)** | **6.26e+00 (1.71e-01)**
> yacht_hydrodynamics + laplace | 2.57e+01 (1.17e+00) | 4.75e+01 (3.51e+00) | **1.64e+01 (3.18e-01)** | **1.61e+01 (2.52e-01)** | **1.62e+01 (2.60e-01)** | **1.67e+01 (3.55e-01)**
> california_housing + laplace | 1.57e+00 (8.25e-02) | 2.06e+00 (1.44e-01) | **1.21e+00 (1.08e-02)** | **1.20e+00 (9.39e-03)** | **1.20e+00 (9.69e-03)** | 1.23e+00 (2.05e-02)
>
> ## NLP
>
> | Data | Standard | Relevance Pursuit | Student-t | Laplace | Huber | Huber + Projection |
> | --- | --- | --- | --- | --- | --- | --- |
> neal + laplace | 1.89e+00 (1.14e-01) | 1.38e+02 (3.98e+01) | **1.57e+00 (9.22e-02)** | **1.55e+00 (9.20e-02)** | **1.55e+00 (9.12e-02)** | **1.55e+00 (9.12e-02)**
> friedman5 + laplace | 4.55e+00 (2.81e-02) | 4.56e+00 (1.88e-02) | 3.64e+00 (4.03e-02) | **3.46e+00 (4.45e-02)** | **3.48e+00 (4.31e-02)** | **3.48e+00 (4.31e-02)**
> friedman10 + laplace | 4.58e+00 (1.25e-02) | 4.57e+00 (1.27e-02) | 3.60e+00 (2.53e-02) | **3.32e+00 (3.01e-02)** | **3.37e+00 (2.61e-02)** | **3.37e+00 (2.61e-02)**
> yacht_hydrodynamics + laplace | **4.74e+00 (4.45e-02)** | 5.30e+00 (8.31e-02) | 4.91e+00 (8.87e-02) | 5.19e+00 (1.17e-01) | 5.12e+00 (1.04e-01) | 9.90e+00 (6.44e-01)
> california_housing + laplace | **1.88e+00 (5.25e-02)** | 2.24e+00 (9.41e-02) | 2.92e+00 (6.01e-02) | 3.33e+00 (7.47e-02) | 3.27e+00 (5.84e-02) | 6.64e+00 (3.28e+00)

---

> ### Author Response · Authors · 2024-08-13
> **A PyTorch implementation of Algikar et.al.'s projection statistics**
>
> Last, we provide the code of the main parts of our translation of the Matlab code of Algikar et.al., 2023 in the following, for you to be able to check its correctness.
> ```
> def projection_statistics(H: Tensor) -> Tensor:
>     """
>
>     Args:
>         H: (n x d)-dim Tensor, i.e. number of data points x dimensionality. NOTE:
>             in the original code, this is taken to be X with an appended ones column.
>
>     Returns:
>         A (n)-dim Tensor of projection statistics.
>     """
>     dtype = H.dtype
>     device = H.device
>     m, n = H.shape
>     M = torch.median(H, dim=0, keepdim=True).values  # (1 x d) row vector
>     u = torch.zeros(m, n, dtype=dtype, device=device)  # i.e. (n x d) matrix
>     v = torch.zeros(m, n, dtype=dtype, device=device)
>     z = torch.zeros(m, 1, dtype=dtype, device=device)  # i.e. (n x 1) matrix
>     P = torch.zeros(m, m, dtype=dtype, device=device)  # i.e. (n x n) matrix
>     eps = 1e-6  # avoiding divide-by-zero issues
>     for kk in range(m):
>         u[kk, :] = H[kk, :] - M  # looping over data points
>         v[kk, :] = u[kk, :] / max(torch.linalg.norm(u[kk, :]), eps)
>         for ii in range(m):
>             z[ii, :] = torch.dot(H[ii, :], v[kk, :])
>         zmed = torch.median(z, dim=0, keepdim=True).values
>         MAD = 1.4826 * (1 + (15 / (m))) * torch.median(torch.abs(z - zmed))
>         for ii in range(m):
>             P[kk, ii] = torch.abs(z[ii] - zmed) / max(MAD, eps)
>
>     PS = torch.amax(P, dim=0)
>     return PS
>
>
> def _compute_projection_statistics_weights(H: Tensor, PS: Tensor) -> Tensor:
>     """This computes the weights for the projection statistics, according to the procedure
>     in the original Matlab code:
>     https://github.com/apooja1/GP-Huber/blob/fee038963b471eb198d59b22d57b89e69f451d8c/Experiments/Friedman
>     """
>     niu = torch.sum(H != 0, dim=-1)
>     cutoff_PS = torch.zeros_like(niu)
>     for i in range(len(niu)):
>         # for chi2, see this post for correspondence of ppf with invcdf:
>         # https://stackoverflow.com/questions/53019080/chi2inv-in-python
>         # the 0.975 was copied from the original matlab code:
>         # https://github.com/apooja1/GP-Huber/blob/fee038963b471eb198d59b22d57b89e69f451d8c/Experiments/Friedman.m#L186
>         cutoff_PS[i] = chi2.ppf(0.975, df=niu[i].item())
>
>     weights = (cutoff_PS / PS.square()).clamp(max=1.0)
>     return weights
>
> def compute_projection_statistics_weights(X: Tensor) -> Tensor:
>     # X: n x d
>     n, d = X.shape
>     H = torch.cat((X, torch.ones(n, 1)), dim=-1)  # n x (d + 1)
>     PS = projection_statistics(H)  # (n x d) -> n
>     return _compute_projection_statistics_weights(H, PS)
> ```
>
> For the Huber loss, we use $\epsilon = 0.45$ and $b = 0.5$, identical to the setting of the [Github repository](https://github.com/apooja1/GP-Huber/blob/fee038963b471eb198d59b22d57b89e69f451d8c/basics/lik_huber.m#L191) of Algikar et.al., 2023.  All reported results were generated with 32 independent replications, reporting the mean and standard error.
>
> An additional implementation detail: the `median` and `max` calls take the extra `dim=0` argument in PyTorch, as the Matlab function are defined to compute the row-wise maximum. Also, we added a small numerical constant `eps` in the code of `projection_statistics` to avoid division-by-zero errors. The rest of the code should be an almost verbatim translation.
>
> **A word of thanks** Thank you again for your valuable suggestions, we believe they make the paper stronger. Let us know if you have additional questions or if all your remaining concerns have been addressed.

---

> > ### Comment · Reviewer_CPiv · 2024-08-14
> >
> > Thank you for providing the performance comparisons with additional models, especially on such short notice. I find the comparison to be very thorough and have adjusted my score accordingly.

---

### Author Rebuttal · Authors · 2024-08-07

We thank the reviewers for their detailed comments and valuable suggestions. We are glad to see that the reviewers found our "novel approach” to be “clearly motivated and presented” and of “broad interest”, and that our work “provides a nice addition to the existing toolbox of methods”.

While reviewers in general found our work “well-written” and “easy to follow”, there were some requests for improving clarity (e.g. Section 4.3), more readable figures, and a more comprehensive discussion of the regression results. We will gladly make these adjustments, and will utilize the additional page to improve legibility and include the additional experiments suggested by the reviewers and added here.

## RCGP Benchmarks
In response to the desire for additional regression results, we ran a comprehensive suite of benchmarks that include various UCI datasets, and included the RCGP method from the contemporaneous work Altamirano et al. (ICML 2024) as an additional baseline, using their publicly available benchmarking suite. The results are presented in Table 1 in the attached pdf. The main takeaways are:
- Overall, RGP-RP outperforms the baselines in terms of Mean Absolute Error (MAE) - the metric chosen by Altamirano et al - in 50% of the test cases, is tied with the standard GP in 40% of cases, and the Student-T GP (t-GP) is best in 10% cases. There is not a single case where RCGP or standard GP outperforms RGP-RP in a statistically significant way (standard errors in parenthesis).

- For uncorrupted data, all methods perform comparably. Notably, RGP-RP shows predictive performance indistinguishable from the standard GP, an indication that the RGP-RP’s Bayesian model selection correctly chooses the outlier-free model.

- For “Uniform” and “Asymmetric” Outliers, as defined in Altamirano et al. 2024, RGP-RP outperforms the baselines for all test cases, showing that our results generalize to more problems.

- RGP-RP takes longer to fit than other baselines but achieves superior predictive performance. Fitting times in these benchmarks are on the order of seconds, and so this should not be a concern for most practical applications. Relative timings that each method took to complete the entire benchmark are:
    - GPR (GPFlow): 1.9x,
    - t-GP (GPFlow): 20.6x
    - RCGPR (GPFlow): 5.7x
    - Standard (BoTorch): 1.0x
    - RGP-RP (BoTorch): 36.2x

_Note: The original RCGP benchmarks from Altamirano et al. (2024) [hard-code the seed that controls the random train-test-splits](https://github.com/maltamiranomontero/RCGP/blob/aff281a39a6be6eefc15d96db8acba6c49224d28/experiments/uci/dataset_api.py#L321), therefore leading the results to only use a single train-test split, contrary to what is indicated in the paper. We edited the code to use different seeds for each replication._

## Twitter Flash Crash
As an additional real-world example, we consider the “twitter flash crash” from Altamirano et al. (2024), see Figure 1 in the attached pdf for results.

_Note: The original notebook sets the RCGP noise variance to the one inferred by GPR, and turns off the learning of the variance. In contrast, RCGP jointly learns the noise variance in the original benchmarks._

In the top row of Figure 1, we show results for
- “RCGP (GPR Noise)”, the original method that forces the variance to the GPR value.
- “RCGP (Trained Noise)” sets the variance to be trainable.
- “RCGP (Fixed Noise)” forces the noise variance to 0.25.

The top row shows that RCGP with the original setting (GPR Noise), and (Fixed Noise) are less affected by the outliers than GPR, but still substantially so. Surprisingly, RCGP (Trained Noise) is even more affected. Further, the RGP-RP (“Relevance Pursuit”) is virtually unaffected by the outlying data points while modeling other high-frequency components of the time series closely.

## Twitter Flash Crash with Additional Training Day

A key shortcoming of RCGP is that its weighting function only uses the distribution of outcomes Y relative to the prior mean $m({\bf x})$. For this to be effective, the outliers need to be separable from the marginal data distribution, like, e.g., for winsorization. The reason RCGP shows improvements on the single-day example is that the outliers at the sell-off time are a-priori separated from most of the data.

If this is not the case, RCGP can up-weight outliers, while down-weighting real data. We illustrate this failure in the bottom row of Figure 1, generated by included data from the preceding day. Here, RCGP is not robust to the outliers - in fact, the weighting function assigns a very large weight to the outlier, while RP-GP exhibits consistently strong performance.

## Additional Results: UCI Data using RP Setup

We also added UCI datasets (yacht, energy, CA housing) into our own benchmarking suite. Figure 2 in the pdf contains the results for the yacht data with “uniform” outliers. Note that RCGP’s “uniform” outlier process for Table 1 is different from ours:

- RCGP’s “uniform” outliers: generated by adding or subtracting a random value uniformly sampled between 3 and 9 standard deviations of the original data, thereby separating outliers from the data.

- RP’s “uniform” outliers: generated by uniformly sampling inside the range of the uncorrupted data.
This leads to the differences between the results reported in Figure 2 and Table 1 on the yacht data.

The results show that RP generally outperforms the baselines in terms of MAE and LL over a range of corruption percentages, and is faster than the robust Student-T and Trimmed-MLL approaches. The results on the CA housing and energy data are qualitatively similar. We will include them in the manuscript.

## An Ask
With these additional results, we believe that the main weaknesses of the paper cited by the reviewers – the regression experiments and comparison to additional baselines (RCGP) – were addressed comprehensively. Based on this, we would like to ask the reviewers to consider raising their score.

---

### Decision · Program_Chairs · 2024-09-25

**Decision:**

Accept (poster)

**Comment:**

All reviewers recommend acceptance. Authors: when preparing the final version of your manuscript, please follow the recommendations and guidance provided by the reviewers. In particular, Section 3 should be improved by including further relevant references from the heteroscedastic GPs literature (e.g., "Heteroscedastic Gaussian Process Regression", "Variational Heteroscedastic Gaussian Process Regression", "Most likely heteroscedastic Gaussian process regression") and discussing how they connect with your method, in particular the lack of an assumption of noise variance smoothly varying with the input.